# PRIOR-FREE TABULAR TEST-TIME ADAPTATION

**Rundong He**[1]    **Jieming Shi**[1*]
[1] Department of Computing, The Hong Kong Polytechnic University
`herundong0@gmail.com, jieming.shi@polyu.edu.hk`

## ABSTRACT

Deep neural networks (DNNs) have been effectively deployed in tabular data modeling for various applications. However, these models suffer severe performance degradation when distribution shifts exist between training and test tabular data. While test-time adaptation (TTA) serves as a promising solution to distribution shifts, existing TTA methods primarily focus on visual modalities and demonstrate poor adaptation when directly applied to tabular modality. Recent efforts have proposed tabular-specific TTA approaches to mitigate distribution shifts on tabular data. Nevertheless, these methods inherently assume the accessibility of source domain or prior and fail to fundamentally address feature shift while overlooking unique characteristics of tabular data, leading to suboptimal adaptation. In this paper, we focus on the problem of *prior-free tabular test-time adaptation* where no access to source data and any prior knowledge is allowed, and we propose a novel method, Prior-Free Tabular Test-Time Adaptation (PFT$_3$A), which has three designs to simultaneously address label shift and feature shift without source domain or prior access. Specially, PFT$_3$A contains the *Class Prior Estimating* module for estimating source-target class priors to calibrate prediction, eliminating dependency on source class prior and mitigating label shift; the *Robust Feature Learning* module for learning robust feature by aligning source-like and target-like features to mitigate feature shift; the *Representative Subspace Exploration* module for eliminating redundant features by projecting feature into subspace to enhance feature alignment. Extensive experiments demonstrate the effectiveness and generalization of PFT$_3$A in tabular TTA tasks. The implementation is at `https://github.com/rundohe/PFT3A`.

## 1 INTRODUCTION

Deep neural networks (DNNs) have achieved remarkable success in modeling tabular data, delivering state-of-the-art performance in critical applications such as fraud detection and healthcare diagnostics (Hollmann et al., 2022). However, their deployment in real-world scenarios is often challenged by distribution shifts of labels and features between training (*source*) and test (*target*) tabular data (Gardner et al., 2023). These shifts, caused by temporal changes, geographic variations, or sampling biases, can lead to significant performance degradation. For example, a model trained on 2024 economic data may perform poorly when predicting 2025 market trends due to shifts in economic policies or geopolitical events that alter underlying data distribution (Okeleke et al., 2024).

Test-time adaptation (TTA) has emerged as a promising solution to address distribution shifts by adapting model parameters using unlabeled test data during inference, without requiring retraining (Liang et al., 2020). While methods like Tent (Wang et al., 2020) and EATA (Niu et al., 2022) have shown success, their development has primarily focused on visual data. However, directly applying these methods to tabular data results in suboptimal performance (Zhou et al., 2024), as they do not consider the distinct characteristics of tabular data.

The Tableshift benchmark (Gardner et al., 2023) identifies two key types of distribution shifts in tabular data: *feature shift*, where feature distributions differ between training and testing, and *label shift*, where label distributions are mismatched. Fig. 1(b) demonstrates that the performance of TabPFN (Hollmann et al., 2025), a tabular foundation model, degrades significantly under distribution

---

*Corresponding author

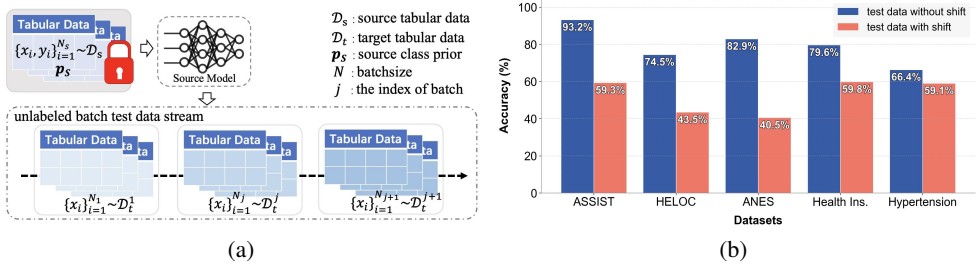

Figure 1: (a) The introduction of prior-free test-time adaptation on tabular data. (b) Performance comparison of the tabular foundation model (TabPFN) on test data with (without) distribution shift.

shifts. Recent tabular-specific TTA methods, like AdapTable (Kim et al., 2024) and TabLog (Ren et al., 2024) address label shift and identify invariant rules for tabular data. Cai & Ye (2025) optimize time-aware data splits and model periodicity via Fourier series to tackle degraded generalization under temporal distribution shifts. Drift-Resilient TabPFN (Helli et al., 2024) integrates causal inference with meta-learning to handle temporal distribution shifts in tabular data. However, these methods assume access to training data, limiting their real-world applicability. FTAT (Zhou et al., 2024) mitigates label shift by optimizing label distributions, addresses feature shift by filtering low-quality predictions, and employs online ensemble learning for robust predictions without requiring source data. While FTAT reduces reliance on training data, it still depends on domain-specific priors. As illustrated in Fig. 1(a), real-world applications frequently require adapting a source-trained model to unlabeled test data with distribution shifts, without access to source data and domain-specific priors.

In this work, we focus on the problem of **prior-free tabular test-time adaptation** where no access to source data and any prior knowledge is allowed, and we propose a novel method, Prior-Free Tabular Test-Time Adaptation ($\text{PFT}_3\text{A}$), which has three designs to simultaneously address label shift and feature shift without source domain access or filtering test data. Specially, $\text{PFT}_3\text{A}$ contains the *Class Prior Estimating* module for estimating source-target class priors to calibrate prediction, eliminating dependency on source class prior and mitigating label shift; the *Robust Feature Learning* module for learning robust feature by minimizing the Kullback-Leibler (KL) divergence between source-like and target-like features to mitigate feature shift; the *Representative Subspace Exploration* module for eliminating redundant features by projecting feature into subspace to enhance feature alignment. To summarize, the contributions of this paper are fourfold:

- We introduce a novel problem, prior-free tabular test-time adaptation, and propose the $\text{PFT}_3\text{A}$ method to address this challenge.
- We design a Class Prior Estimating module to mitigate label shift by estimating source-target class priors, eliminating dependency on source class prior.
- We enhance feature alignment through Robust Feature Learning and Representative Subspace Exploration, addressing feature shift and eliminating redundant features.
- Extensive experiments demonstrate that $\text{PFT}_3\text{A}$ outperforms state-of-the-art TTA methods.

## 2   RELATED WORK

**Test-Time Adaptation (TTA).** TTA adapts a pre-trained source model to distribution shifts in test data without accessing source data. Existing methods primarily address feature shifts, where conditional distributions remain consistent but marginal distributions differ. Approaches include leveraging class prototypes or pseudo-labels (Chen et al., 2022; Jang et al., 2022; Goyal et al., 2022; Liang et al., 2020), and employing self-supervised learning (SSL) on auxiliary tasks like rotation prediction (Liang et al., 2021) or masked autoencoding (Gandelsman et al., 2022; Ren et al., 2023) to enhance robustness. Test-Time Training methods, such as TTT(Sun et al., 2020) and TTT+(Liu et al., 2021), jointly train models on primary and auxiliary SSL tasks, enabling updates with unlabeled test data, but its reliance on source data limits applicability. Fully TTA methods, such as Tent (Wang et al., 2020) and EATA (Niu et al., 2022), adapt models without source data by updating batch normalization (BN) layers or employing sample selection strategies. Similar BN

adaptation techniques are explored in (Goyal et al., 2022). TDA (Karmanov et al., 2024) enables effective and efficient test-time adaptation with vision-language models. While effective in vision tasks, these methods remain largely unexplored for tabular data.

Recent works like AdapTable (Kim et al., 2024) and TabLog (Ren et al., 2024) address label shifts and identify invariant rules for tabular data. Cai & Ye (2025) optimize time-aware data splits and model periodicity via Fourier series to tackle degraded generalization under temporal distribution shifts. Drift-Resilient TabPFN (Helli et al., 2024) integrates causal inference with meta-learning to handle temporal distribution shifts in tabular data. However, these methods assume access to training data, limiting their real-world applicability. FTAT (Zhou et al., 2024) mitigates label shift by optimizing label distributions, addresses feature shift by filtering low-quality predictions, and employs online ensemble learning for robust predictions without requiring source data. While FTAT reduces reliance on training data, it still depends on domain-specific priors. In contrast, we propose a prior-agnostic test-time adaptation framework for tabular data, eliminating dependencies on training data and domain-specific priors.

**Deep Tabular Learning.** Learning on tabular data focuses on applying advanced techniques to tasks like classification and regression. Unlike image or text data, which exhibit spatial or sequential patterns, tabular data is often high-dimensional and heterogeneous, posing unique challenges for traditional models. Recent works have introduced specialized deep learning architectures to address these challenges. FT-Transformer (Gorishniy et al., 2021) employs a feature tokenizer to process diverse feature columns and generate optimal embeddings. TabTransformer (Huang et al., 2020) leverages self-attention to model complex feature interactions, while TabNet (Arik & Pfister, 2021) integrates decision trees with deep learning for enhanced interpretability. Other methods, such as Deep Feature Interaction Network (DeepFIN) (Klambauer et al., 2017) and TabCNN (Gorishniy et al., 2022), further explore effective representation learning for tabular data. TabPFN (Hollmann et al., 2022; 2025), a transformer-based foundation model, excels in supervised classification on small tabular datasets by using in-context learning and a causal prior for efficient Bayesian inference. However, despite their success in i.i.d. settings, these models often struggle under distribution shifts.

## 3 PROBLEM AND ANALYSIS

In this section, we define the prior-free tabular test-time adaptation problem, introducing its notations and formulation (Section 3.1). We then identify the core challenges of tabular TTA and highlight the limitations of existing methods through three focused analyses (Section 3.2).

### 3.1 PROBLEM FORMULATION

**Definition 1** *(Test-time Adaptation (TTA).) Let $D_s = \{(x_i, y_i)\}_{i=1}^{N_s}$ be $N_s$ labeled samples from a source distribution $Q$ over $\mathcal{X} \times \mathcal{Y}$, where $\mathcal{X}$ is the input space and $\mathcal{Y} = \{1, \ldots, K\}$ is the label space with $K$ classes. During adaptation, unlabeled target data $D_t = \{x_i\}_{i=1}^{N_t}$, sampled from a different distribution $P$ ($P \neq Q$), arrives sequentially in batch $D_t^j$. Given a source model $f_{\theta_0} : \mathcal{X} \to \mathcal{Y}$ trained on $D_s$ with initial parameters $\theta_0$, TTA dynamically updates $f_{\theta_0}$ using only the incoming unlabeled batch $D_t^j$, without accessing $D_s$, to address distribution shifts.*

**Definition 2** *(Prior-free Tabular Test-time Adaptation.) Extending Definition 1, this setting focuses on tabular data, where both $D_s$ and $D_t$ belong to the tabular modality. Given a source model $f_{\theta_0} : \mathcal{X} \to \mathcal{Y}$ trained on $D_s$ with initial parameters $\theta_0$, the goal is to adapt $f_{\theta_0}$ using only the incoming unlabeled tabular batches $D_t^j$, without access to $D_s$ or source class prior $p_S$ (i.e., prior-free), to address distribution shifts. Here, $\mathcal{X} \subseteq \mathbb{R}^d$ represents the tabular input space, where $d$ denotes the number of columns, each being either continuous or discrete.*

### 3.2 ANALYSIS

Feature shift and label shift are two common types of distribution shifts in tabular data (Gardner et al., 2023). Feature shift arises when the feature distributions between training and testing data differ. For instance, in ASSIST (Gardner et al., 2023), cross-school generalization suffers accuracy drop due to disparities in input feature distributions (e.g., problem-solving behaviors, study habits)

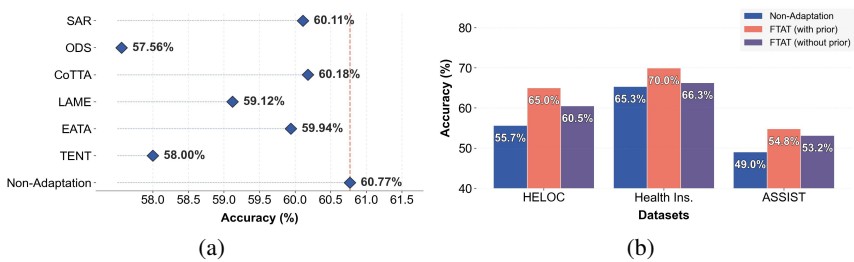

Figure 2: (a) The average accuracy results of various classic TTA methods and Non-Adaptation on five tabular datasets.(b) Accuracy Comparison of Non-Adaptation, FTAT with prior, and FTAT without prior across HELOC, Health Ins., and ASSIST.

across schools. Label shift occurs when the label distribution varies between domains. Existing TTA methods struggle with tabular data. Fig. 2(a) compares the average accuracy of classic TTA methods, including TENT (Wang et al., 2020), EATA (Niu et al., 2022), CoTTA (Wang et al., 2022), SAR (Niu et al., 2023), LAME (Boudiaf et al., 2022), ODS (Zhou et al., 2023), and Non-Adaptation (direct evaluation without TTA) on five tabular datasets. These methods often underperform Non-Adaptation, with average accuracies of 58% (TENT) and 57.56% (ODS) compared to 60.77% for Non-Adaptation. Originally developed for visual data, these methods struggle with tabular TTA. Moreover, existing tabular TTA methods face the following three major limitations: (i) *Ineffectiveness in prior-free scenarios.* Current methods like AdapTable (Kim et al., 2024) and TabLog (Ren et al., 2024) address label and covariate shifts but rely on access to source data, which is often unavailable in real-world applications. While FTAT (Zhou et al., 2024) eliminates the need for source data, it still depends on class priors derived from the source distribution, limiting its effectiveness in prior-free settings. Fig. 2(b) shows that the absence of class priors leads to a significant accuracy drop for FTAT across datasets such as HELOC, Health Ins., and ASSIST. This underscores the need for TTA methods that are both source-free and prior-free for robust adaptation in real-world tabular scenarios. (ii) *Limitations of filtering low-confidence samples.* Feature shift arises from discrepancies between source and target feature distributions. Filtering low-confidence target samples excludes target-specific data without addressing the underlying mismatch. High-confidence samples often resemble the source domain, while low-confidence ones capture target-specific traits. Discarding the latter biases the model towards source-like representations, limiting generalization. Feature alignment, instead, bridges the source-target gap by learning invariant features, effectively mitigating feature shift (Chen et al., 2019). (iii) *Over-alignment of all feature dimensions.* In tabular tasks with limited class diversity (e.g., binary classification in Tableshift (Gardner et al., 2023)), empirical findings (Zhang et al., 2023) highlight significant feature space redundancy, where only a subset of features is predictive. Over-aggressive feature alignment leads to two key issues: reliance on redundant, non-discriminative features, fostering spurious correlations; and neglect of critical discriminative features, degrading adaptation performance. Selectively aligning discriminative features, as demonstrated in Table 10, improves performance across datasets.

## 4 METHODOLOGY

In this section, we introduce our proposed method, Prior-Free Tabular Test-Time Adaptation (PFT$_3$A), which addresses the major challenges in tabular TTA analyzed in Section 3.2. PFT$_3$A consists of three modules, as shown in Figure 3: (1) Class Prior Estimating, which estimates the class priors of the source and target domains by identifying source-like and target-like data from unlabeled batched target data, without requiring access to source data or class priors; (2) Robust Feature Learning, which mitigates feature shift by aligning the feature distributions between the source-like and target-like data, without relying on filtering low-confidence predictions; (3) Representative Subspace Exploration, which identifies and aligns representative subspaces using subspace detection methods, leveraging the unique characteristics of tabular data and the Tableshift benchmark, without aligning all feature dimensions. In what follows, we will provide a detailed introduction to these three modules.

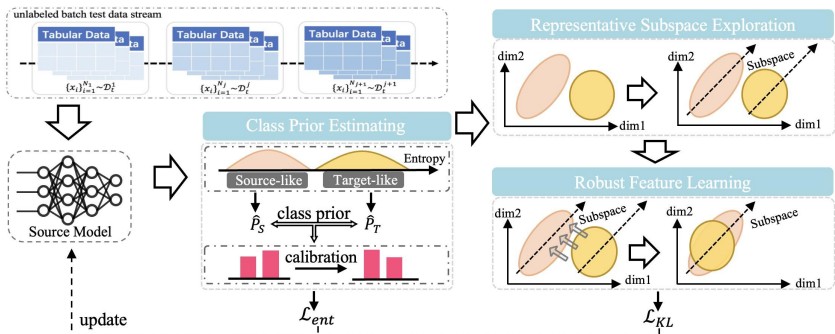

Figure 3: The framework PFT₃A for prior-free tabular test-time adaption. It contains the Class Prior Estimating for estimating source-target class priors to calibrate prediction, Robust Feature Learning for aligning feature to mitigate feature shift, and Representative Subspace Exploration for eliminating redundant features to enhance feature alignment.

## 4.1 CLASS PRIOR ESTIMATING

As analyzed in Section 3.2, existing TTA methods that use class priors on tabular data suffer from degraded performance in the prior-free setting. In the prior-free TTA in Definition 2, the source model $f_{\theta_0}$ (trained on source data $D_s$) should adapt to unlabeled target data $D_t$ without access to any prior knowledge (e.g., source data $D_s$ and class priors).

Aiming to estimate the class priors, we propose Class Prior Estimating module during testing time. The main challenge here is how to estimate the class priors of the source and target domains without access to the source data. Our idea is to leverage the model trained on the source domain to identify the source-like and target-like data from the unlabeled target data. Specifically, the model trained on the source domain tends to be more confident when encountering samples resembling the source domain. Thus, we treat samples with low prediction uncertainty as *source-like* examples to approximate the source proxy distribution, and samples with high prediction uncertainty as *target-like* examples to approximate the target proxy distribution (Section 4.1.1). Then we estimate the class priors of the source and target domains based on the identified source-like and target-like data and calibrate the model's predictions to mitigate label shift during test-time adaptation (TTA) (Section 4.1.2).

### 4.1.1 ESTIMATING PROXY DISTRIBUTIONS

Firstly, we feed the data $\mathbf{x}_i$ in the $j$-th unlabeled batched target tabular data $D_t^j$ into the model $f_{\theta_{j-1}}$ to obtain the predicted probabilities $\hat{\mathbf{p}}_i(\mathbf{x}_i, f_{\theta_{j-1}})$ by

$$\hat{\mathbf{p}}_i = f_{\theta_{j-1}}(\mathbf{x}_i) = \left[\hat{p}_i^{(1)}, \hat{p}_i^{(2)}, \ldots, \hat{p}_i^{(K)}\right], \mathbf{x}_i \in D_t^j, i \in \{(j-1)b, (j-1)b+1, \ldots, jb\}, \quad (1)$$

where $b$ denotes the size of batch, $\hat{p}_i^{(k)}$ denotes the predicted probability that example $\mathbf{x}_i$ belongs to $k$-th class ($k = 1, 2, \ldots, K$), $K$ denotes the number of category.

We use entropy to quantify the prediction uncertainty of the model. The entropy of the probabilities $\hat{\mathbf{p}}_i$ is defined below. Following Zhou et al. (2024), data arrives in an online fashion, processed in batches.

$$H(\hat{\mathbf{p}}_i) = -\sum_{k=1}^K \hat{p}_i^{(k)} \log \hat{p}_i^{(k)} . \quad (2)$$

For $j$-th batch data $D_t^j$, we mine source-like set $\hat{S}^j$ and target-like set $\hat{T}^j$ by

$$\hat{S}^j = \{\mathbf{x}_i | \mathbf{x}_i \in D_t^j, H(\hat{\mathbf{p}}_i) < \epsilon\}, \hat{T}^j = \{\mathbf{x}_i | \mathbf{x}_i \in D_t^j, H(\hat{\mathbf{p}}_i) > \epsilon\}, \quad (3)$$

where $\epsilon$ denotes a threshold and $j$ denotes the index of the $j$-th batch.

### 4.1.2 ESTIMATING THE CLASS PRIORS

We initially estimate the class priors of the source and target domains based on $\hat{S}^1$ and $\hat{T}^1$ in the first unlabeled batched data $D_t^1$. Specifically, we estimate class prior $\hat{\mathbf{p}}_S$ of source domain by

$$\hat{\mathbf{p}}_S = \frac{1}{N_{\hat{S}^1}} \sum_{i \in \hat{S}^1} \hat{\mathbf{p}}_i , \quad (4)$$

where $N_{\hat{S}^1}$ denotes the number of examples in the source-like set $\hat{S}^1$. Similarly, target domain's class prior $\hat{\mathbf{p}}_T$ can be estimated initially by

$$\hat{\mathbf{p}}_T = \tfrac{1}{N_{\hat{T}^1}} \sum_{i \in \hat{T}^1} \hat{\mathbf{p}}_i \,, \tag{5}$$

where $N_{\hat{T}^1}$ denotes the number of examples in the first target-like set $\hat{T}^1$.

We use only the first unlabeled batch from the target domain to obtain the source-like set. This is because the source model, trained solely on source data, better encapsulates the source domain's knowledge before being updated with target data. Since the source model has not encountered target data, its predictions on target-like data may lack accuracy, leading to biased estimates of the target domain's class priors. To mitigate this bias, we iteratively refine the target domain's class priors:

$$\hat{\mathbf{p}}_T^j = \text{Norm}(\hat{\mathbf{p}}_T^{j-1} - \hat{C}_j^{-1} \tilde{\mathbf{p}}_S^j) \,, \tag{6}$$

where $\hat{C}_j^{-1} \tilde{\mathbf{p}}_S^j$ represents the estimated bias correction term derived from the $j$-th batch data $D_t^j$, $\hat{C}_j$ is the covariate matrix computed on $D_t^j$, and Norm() ensures the resulting distribution sums to one. $\tilde{\mathbf{p}}_S^j$ is defined as:

$$\tilde{\mathbf{p}}_S^j = \tfrac{1}{N_{\hat{S}^j}} \sum_{i \in \hat{S}^j} \hat{\mathbf{p}}_i \,. \tag{7}$$

Using the estimated class priors $\hat{\mathbf{p}}_S$ and $\hat{\mathbf{p}}_T$ for the source and target domains, we calibrate the model's predictions to address label shift during test-time adaptation. Specifically, for an example $\mathbf{x}_i$ in the $j$-th batch $D_t^j$, the model's prediction $f_{\theta_{j-1}}(\mathbf{x}_i)$ is adjusted to obtain the calibrated prediction $\tilde{f}_{\theta_{j-1}}(\mathbf{x}_i)$ as:

$$\tilde{f}_{\theta_{j-1}}(\mathbf{x}_i) = f_{\theta_{j-1}}(\mathbf{x}_i) \circ \frac{\hat{\mathbf{p}}_T^j}{\hat{\mathbf{p}}_S} \,, \tag{8}$$

where $f_{\theta_{j-1}}(\mathbf{x}_i)$ is the raw model prediction for the data point $\mathbf{x}_i$ in $j$-th batched data $D_t^j$.

## 4.2 ROBUST FEATURE LEARNING

In this section, we introduce the Robust Feature Learning (RFL) module, which aims to learn robust features to mitigate feature shift.

From Eq. 3, we derive the source-like set $\hat{S}^j$ and target-like set $\hat{T}^j$ for the $j$-th batch $D_t^j$. By passing $\hat{S}^j$ and $\hat{T}^j$ through the feature extractor $g$, we obtain the source-like feature set $g(\hat{S}^j)$ and target-like feature set $g(\hat{T}^j)$. Assuming the feature distributions follow a Gaussian distribution Adachi et al. (2024), we estimate the proxy source and target feature distributions based on $g(\hat{S}^j)$ and $g(\hat{T}^j)$. The mean $\boldsymbol{\mu}_S^j$ and variance $(\boldsymbol{\sigma}^2)_S^j$ of the proxy source feature distribution are computed as:

$$\boldsymbol{\mu}_S^j = \frac{1}{N_{\hat{S}^j}} \sum_{\mathbf{x}_i \in \hat{S}^j} \mathbf{z}_i, \quad (\boldsymbol{\sigma}^2)_S^j = \frac{1}{N_{\hat{S}^j}} \sum_{\mathbf{x}_i \in \hat{S}^j} (\mathbf{z}_i - \boldsymbol{\mu}_S^j) \odot (\mathbf{z}_i - \boldsymbol{\mu}_S^j) \,, \tag{9}$$

where $\mathbf{z}_i = g_{\phi_{j-1}}(\mathbf{x}_i) \in \mathbb{R}^d$, $\phi$ denotes the parameters of feature extractor $g$, $d$ denotes the dimension of feature, and $\odot$ denotes the element-wise product.

Similarly, we compute the mean $\boldsymbol{\mu}_T^j$ and variance $(\boldsymbol{\sigma}^2)_T^j$ of proxy source feature distribution by

$$\boldsymbol{\mu}_T^j = \frac{1}{N_{\hat{T}^j}} \sum_{\mathbf{x}_i \in \hat{T}^j} \mathbf{z}_i, \quad (\boldsymbol{\sigma}^2)_T^j = \frac{1}{N_{\hat{T}^j}} \sum_{\mathbf{x}_i \in \hat{T}^j} (\mathbf{z}_i - \boldsymbol{\mu}_T^j) \odot (\mathbf{z}_i - \boldsymbol{\mu}_T^j) \,. \tag{10}$$

Then, we align feature distributions to learn robust feature by minimizing the Kullback-Leibler (KL) divergence between proxy source feature distribution and proxy target feature distribution:

$$\begin{aligned}
\mathcal{L}_{KL} &= \text{KL}\left(\mathcal{N}(\boldsymbol{\mu}_S^j, (\boldsymbol{\sigma}^2)_S^j) \,\|\, \mathcal{N}(\boldsymbol{\mu}_T^j, (\boldsymbol{\sigma}^2)_T^j)\right) \\
&= [\log((\boldsymbol{\sigma}^2)_T^j/(\boldsymbol{\sigma}^2)_S^j) + \{(\boldsymbol{\mu}_S^j - \boldsymbol{\mu}_T^j)^2 + (\boldsymbol{\sigma}^2)_S^j\}/(\boldsymbol{\sigma}^2)_T^j - 1]/2 \,.
\end{aligned} \tag{11}$$

In the Tableshift benchmark (Cai & Ye, 2025), which focuses on binary classification, the features are often less diverse compared to multi-class classification. They are typically distributed within a

small subspace of the feature space, as many dimensions exhibit zero variance (Zhang et al., 2023; Adachi et al., 2024). This dimensional sparsity poses two key challenges for feature alignment: 1) The computation of KL divergence in Eq. 11 becomes numerically unstable due to the inclusion of variance terms from underutilized dimensions; 2) The alignment effectiveness is reduced, as most dimensions contribute negligibly to the representative subspace. To mitigate these issues, we propose to explore the representative subspace of the feature space, which is detailed in the next section.

## 4.3 REPRESENTATIVE SUBSPACE EXPLORATION

As analyzed in Section 3.2, aligning features across the full feature space can be suboptimal, as not all dimensions contribute meaningfully to tabular data TTA. To address this, we focus on identifying and aligning the most informative subspace that captures the key dimensions relevant for adaptation. Specifically, we employ a subspace detection method inspired by Principal Component Analysis to explore the representative subspace for prior-free tabular TTA. First, we compute the covariance matrix for the proxy source features as:

$$\Sigma_S^j = \frac{1}{N_{\hat{S}^j}} \sum_{\mathbf{x}_i \in \hat{S}^j} (\hat{\mathbf{z}}_i - \hat{\boldsymbol{\mu}}_S^j)(\hat{\mathbf{z}}_i - \hat{\boldsymbol{\mu}}_S^j)^{\mathrm{T}}, \qquad (12)$$

where the proxy source mean $\hat{\boldsymbol{\mu}}_S^j$ is the same as Eq. 9. The subspace is spanned by the eigenvectors of the covariance matrix $\Sigma_S^j$, denoted by $\mathbf{V}_S$ ($\mathbf{V}_S = [\mathbf{v}_S^1, \mathbf{v}_S^2, \cdots]$). The corresponding eigenvalues $\boldsymbol{\lambda}_S$ ($\boldsymbol{\lambda}_S = [\lambda_S^1, \lambda_S^2, \cdots]$) represent the variance of the source features along the direction $\mathbf{V}_S$. We use the top-$m$ largest eigenvalues $\lambda_S^1, \lambda_S^2, \cdots, \lambda_S^m$, the corresponding eigenvectors $\mathbf{v}_S^1, \mathbf{v}_S^2, \cdots, \mathbf{v}_S^m$.

We project the feature $g_{\phi_{j-1}}(\mathbf{x}_i)$ of example $\mathbf{x}_i$ into the subspace by

$$\mathbf{z}_i^{\mathrm{proj}} = \mathbf{V}_S g_{\phi_{j-1}}(\mathbf{x}_i), \qquad (13)$$

where $\mathbf{z}_i^{\mathrm{proj}} \in \mathbb{R}^m$, $\mathbf{V}_S = [\mathbf{v}_S^1, \mathbf{v}_S^2, \cdots, \mathbf{v}_S^m] \in \mathbb{R}^{m \times d}$.

Then, we compute the projected mean $\hat{\boldsymbol{\mu}}_S^j$ and variance $(\hat{\boldsymbol{\sigma}}^2)_S^j$ of proxy source domain according to Eq. 9 and compute the projected mean $\hat{\boldsymbol{\mu}}_T^j$ and variance $(\hat{\boldsymbol{\sigma}}^2)_T^j$ of proxy target domain according to Eq. 10. After that, we align the projected feature distributions of the proxy source and target domains in the subspace by

$$\mathcal{L}_{KL} = \mathrm{KL}\left(\mathcal{N}(\hat{\boldsymbol{\mu}}_S^j, (\hat{\boldsymbol{\sigma}}^2)_S^j) \,\|\, \mathcal{N}(\hat{\boldsymbol{\mu}}_T^j, (\hat{\boldsymbol{\sigma}}^2)_T^j)\right). \qquad (14)$$

Moreover, similar to (Zhou et al., 2024), we employ entropy minimization loss in accordance with classical TTA methods, defined by

$$\mathcal{L}_{ent} = -\sum_{k=1}^K \hat{p}_i^{(k)} \log \hat{p}_i^{(k)}, \qquad (15)$$

where $\hat{p}_i^{(k)}$ denotes the predicted probability that example $\mathbf{x}_i$ belongs to class $k$ ($k = 1, 2, \ldots, K$), $K$ denotes the number of category.

Jointly considering Eq. 14 and Eq. 15, the final optimization objective is defined as follows,

$$\mathcal{L}_{all} = \beta_1 \mathcal{L}_{KL} + \beta_2 \mathcal{L}_{ent}. \qquad (16)$$

By minimizing Eq. 16, we can effectively mitigate label shift and feature shift, enabling tabular test-time adaptation under prior-free scenarios.

Assuming Gaussian distributions in Eq. 14 is reasonable for the following reasons: First, we can easily compute the KL divergence in a closed form by assuming the Gaussian distribution. Second, features follow a Gaussian-like distribution when projected onto the feature subspace under subspace. This is due to the central limit theorem, i.e., the features are more likely to follow a Gaussian distribution in the subspace as the number of the original feature dimensions increases. Moreover, since our method uses the PCA, the features projected onto the subspace are decorrelated. KL divergence provides an information-theoretic measure of distribution discrepancy, which aligns with our goal of minimizing the information loss when adapting between domains.

Table 1: Performance comparison with TabTransformer as backbone. The best is in bold.

| Method | HELOC | | | ANES | | | Health Ins. | | | ASSIST | | | Hypertension | | | Avg | | |
|---|---|---|---|---|---|---|---|---|---|---|---|---|---|---|---|---|---|---|
| | Acc. | BAcc. | F1 | Acc. | BAcc. | F1 | Acc. | BAcc. | F1 | Acc. | BAcc. | F1 | Acc. | BAcc. | F1 | Acc. | BAcc. | F1 |
| Non-Adaptation | 55.66 | 59.60 | 44.30 | 78.95 | 75.30 | 84.23 | 65.35 | 70.11 | 65.88 | 49.04 | 53.07 | 58.92 | 54.87 | 59.24 | 46.25 | 60.77 | 63.46 | 59.92 |
| TENT | 56.37 | 59.60 | 50.34 | 78.82 | 75.18 | 84.14 | 65.22 | 70.07 | 65.62 | 47.83 | 53.11 | 61.27 | 41.74 | 50.12 | 0.78 | 58.00 | 61.62 | 52.43 |
| EATA | 56.37 | 57.04 | 52.57 | 78.82 | 75.18 | 84.14 | 65.35 | 70.11 | 65.88 | 45.28 | 51.44 | 61.40 | 54.86 | 59.24 | 46.24 | 59.94 | 62.60 | 62.05 |
| LAME | 51.71 | 59.56 | 44.30 | 78.57 | 74.77 | 84.04 | 65.35 | 70.11 | 65.88 | 45.12 | 51.30 | 61.40 | 54.87 | 59.24 | 46.25 | 59.12 | 63.00 | 60.37 |
| CoTTA | 56.37 | 59.60 | 50.34 | 78.82 | 75.18 | 84.14 | 65.35 | 70.11 | 65.88 | 45.51 | 51.64 | **61.56** | 54.87 | 59.24 | 46.25 | 60.18 | 63.15 | 61.63 |
| ODS | 52.19 | 56.77 | 44.70 | 78.48 | 74.69 | 83.97 | 57.14 | 64.75 | 51.33 | 45.12 | 51.30 | 61.40 | 54.87 | 59.24 | 46.25 | 57.56 | 61.35 | 57.53 |
| SAR | 56.37 | 59.60 | 50.34 | 78.82 | 75.18 | 84.14 | 65.35 | 70.11 | 65.88 | 45.12 | 51.30 | 61.40 | 54.87 | 59.24 | 46.25 | 60.11 | 63.09 | 61.60 |
| FTAT | 60.54 | 63.22 | 55.81 | 79.46 | 76.29 | **84.34** | 66.31 | 70.84 | 67.19 | 53.17 | 55.85 | 58.94 | 61.78 | **61.65** | 65.62 | 64.25 | 65.57 | 66.38 |
| PFT$_3$A (ours) | **66.17** | **65.17** | **70.91** | **80.33** | **78.72** | 84.06 | **74.13** | **72.67** | **79.33** | **59.29** | **59.15** | 59.73 | **63.03** | 61.39 | **69.20** | **68.59** | **67.42** | **72.65** |

Table 2: Performance comparison with MLP as backbone. The best is in bold.

| Method | HELOC | | | ANES | | | Health Ins. | | | ASSIST | | | Hypertension | | | Avg | | |
|---|---|---|---|---|---|---|---|---|---|---|---|---|---|---|---|---|---|---|
| | Acc. | BAcc. | F1 | Acc. | BAcc. | F1 | Acc. | BAcc. | F1 | Acc. | BAcc. | F1 | Acc. | BAcc. | F1 | Acc. | BAcc. | F1 |
| Non-Adaptation | 54.37 | 58.25 | 40.02 | 79.11 | 75.66 | 84.24 | 65.79 | 70.68 | 66.21 | 55.86 | 60.81 | 66.42 | 58.76 | 61.69 | 55.46 | 62.78 | 65.42 | 62.47 |
| TENT | 54.35 | 58.24 | 39.95 | 78.07 | 74.09 | 83.76 | 64.30 | 69.79 | 63.87 | 50.87 | 56.41 | 63.99 | 41.67 | 50.07 | 0.49 | 57.85 | 61.72 | 50.41 |
| EATA | 54.37 | 58.25 | 40.02 | 78.13 | 74.20 | 83.79 | 65.78 | 70.68 | 66.21 | 55.86 | 60.81 | 66.42 | 57.81 | 61.19 | 52.87 | 62.39 | 65.03 | 61.86 |
| LAME | 43.10 | 50.00 | 30.10 | 63.50 | 54.60 | 46.80 | 63.44 | 69.14 | 62.61 | 45.12 | 51.30 | 61.40 | 58.63 | 61.64 | 55.12 | 54.76 | 57.34 | 51.21 |
| CoTTA | 54.36 | 58.25 | 40.03 | 78.13 | 74.20 | 83.79 | 65.79 | 70.68 | 66.21 | 55.86 | 60.81 | 66.42 | 58.76 | 61.69 | 55.46 | 62.58 | 65.13 | 62.38 |
| ODS | 43.10 | 50.00 | 30.10 | 63.50 | 54.60 | 46.80 | 63.45 | 69.14 | 62.62 | 45.12 | 51.30 | 61.40 | 57.12 | 60.80 | 51.41 | 54.46 | 57.17 | 50.47 |
| SAR | 52.32 | 56.74 | 33.16 | 78.13 | 74.20 | 83.79 | 65.79 | 70.68 | 66.21 | 55.86 | 60.81 | 66.42 | 58.21 | 61.50 | 53.81 | 62.06 | 64.79 | 60.68 |
| FTAT | 60.89 | 62.67 | 59.14 | 79.29 | 75.83 | **84.42** | 67.25 | 71.60 | 68.38 | 51.84 | 57.26 | 64.45 | 63.28 | **63.18** | 66.99 | 64.51 | 66.11 | 68.68 |
| PFT$_3$A (ours) | **65.26** | **62.87** | **72.43** | **79.84** | **79.23** | 82.94 | **73.09** | **73.63** | **77.20** | **63.54** | **65.57** | **66.45** | **64.53** | 61.83 | **71.93** | **69.25** | **68.63** | **74.19** |

# 5 EXPERIMENTS

We begin by detailing the experimental setup, followed by a comparative evaluation of PFT$_3$A against existing TTA methods on five benchmark datasets. Finally, we present an ablation study and analyze the results, emphasizing the strengths and insights of our approach.

## 5.1 EXPERIMENTAL SETUP

**Datasets.** Following FTAT (Zhou et al., 2024), we evaluate our prior-free TTA approach on five benchmark datasets from the TableShift benchmark Gardner et al. (2023): HELOC, ANES, ASSIST, Hypertension, and Health Ins. These datasets exhibit significant distribusion shifts, with sample sizes ranging from 10K to 5M and feature dimensions varying from 26 to 365. This diversity enables a comprehensive evaluation of our method's robustness across a wide range of tabular data scenarios under distribution shifts.

**Compared Methods.** We first compare PFT$_3$A with the non-adapted source model, referred to as Non-Adaptation. Then, we compare PFT$_3$A with various TTA methods, including typical TTA methods (i.e., TENT (Wang et al., 2020) and EATA (Niu et al., 2022)), continual TTA methods (i.e., CoTTA (Wang et al., 2022)), robust FTTA methods (i.e., SAR (Niu et al., 2023), LAME (Boudiaf et al., 2022), ODS (Zhou et al., 2023)), and tabular TTA method (i.e., FTAT (Zhou et al., 2024).

**Backbone Models.** We use three representative deep tabular models: MLP, TabTransformer (Huang et al., 2020) and FT-Transformer (Gorishniy et al., 2021) as the backbone.

**Metrics.** Following FTAT (Zhou et al., 2024), we use accuracy (Acc), balanced accuracy (BAcc), and F1 score to evaluate the performance. More experimental details and analyses can be found in the Appendix.

## 5.2 MAIN RESULTS

Tables 6, 8, and 9 report the performance of PFT$_3$A and baselines on the five datasets using TabTransformer, MLP, and FT-Transformer as backbones, respectively. We observe that PFT$_3$A consistently outperforms the baselines under most metrics across all datasets. For example, as shown in Table 6, PFT$_3$A improves avg Acc, avg BAcc, and avg F1 by 7.82%, 3.96%, and 12.73%, respectively, compared to Non-Adaptation, demonstrating the necessity of adapting the model on unlabeled test data to mitigate distribution shifts effectively. Compared to typical TTA methods (e.g., EATA), PFT$_3$A

Table 3: Performance comparison with FT-Transformer as backbone. The best is in bold.

| Method | HELOC | | | ANES | | | Health Ins. | | | ASSIST | | | Hypertension | | | Avg | | |
|---|---|---|---|---|---|---|---|---|---|---|---|---|---|---|---|---|---|---|
| | Acc. | BAcc. | F1 | Acc. | BAcc. | F1 | Acc. | BAcc. | F1 | Acc. | BAcc. | F1 | Acc. | BAcc. | F1 | Acc. | BAcc. | F1 |
| Non-Adaptation | 46.26 | 52.48 | 13.32 | 75.47 | 71.50 | **81.80** | 58.33 | 65.44 | 54.06 | 58.32 | 62.99 | **67.63** | 58.88 | 61.84 | 55.71 | 59.45 | 62.85 | 54.50 |
| TENT | 44.98 | 51.45 | 8.11 | 63.02 | 54.52 | 76.19 | 36.44 | 50.05 | 0.24 | 58.25 | 62.91 | 67.57 | 47.01 | 53.88 | 18.83 | 54.05 | 57.84 | 48.68 |
| EATA | 45.95 | 52.23 | 12.27 | 74.65 | 70.16 | 81.51 | 57.40 | 64.86 | 52.23 | 48.04 | 53.85 | 62.60 | 58.84 | 61.82 | 55.62 | 57.23 | 60.24 | 52.85 |
| LAME | 43.14 | 50.03 | 0.20 | 75.37 | 71.35 | 81.73 | 59.08 | 65.44 | 55.91 | 56.54 | 62.98 | **67.63** | 58.78 | 61.78 | 55.47 | 59.40 | 62.06 | 51.16 |
| CoTTA | 46.26 | 52.48 | 10.67 | 75.47 | 71.50 | **81.80** | 58.33 | 65.44 | 54.06 | 58.25 | 62.91 | 67.57 | 58.88 | 61.84 | 55.71 | 59.44 | 62.83 | 53.96 |
| ODS | 43.14 | 50.03 | 0.20 | 75.41 | 71.41 | 81.75 | 59.99 | 65.54 | 58.37 | 57.39 | 62.16 | 67.14 | 58.77 | 61.78 | 55.45 | 58.94 | 62.18 | 52.58 |
| SAR | 43.30 | 50.20 | 30.60 | 75.47 | 71.50 | **81.80** | 58.33 | 65.44 | 54.06 | 58.25 | 62.91 | 67.57 | 59.64 | 62.24 | 57.52 | 59.00 | 62.46 | 58.31 |
| FTAT | 59.20 | 61.51 | 55.52 | 76.06 | 73.29 | 81.38 | 66.45 | 70.50 | 67.85 | 58.32 | 62.95 | 67.56 | 62.34 | **63.30** | 64.16 | 64.47 | 66.31 | 67.29 |
| PFT₃A (ours) | **65.46** | **62.70** | **73.17** | **76.62** | **75.42** | 80.60 | **73.88** | **72.84** | **78.87** | **59.36** | **63.53** | 67.43 | **64.74** | 63.07 | **70.74** | **68.01** | **67.51** | **74.16** |

achieves improvements of 8.65%, 4.82%, and 10.6% in avg Acc, avg BAcc, and avg F1, respectively, highlighting the limitations of directly applying typical TTA methods to tabular data, which often underperform Non-Adaptation. Similarly, PFT₃A outperforms continual TTA methods (e.g., CoTTA) by 8.41%, 4.27%, and 11.02% in the same metrics, indicating that naive application of continual TTA methods to tabular data fails to yield significant improvements. Robust TTA methods (e.g., ODS) achieve avg Acc, avg BAcc, and avg F1 scores of 57.56%, 61.35%, and 57.53%, respectively, which are lower than all other approaches, confirming their ineffectiveness for tabular TTA. FTAT, a tabular-specific TTA method, achieves 64.25%, 65.57%, and 66.38% in avg Acc, avg BAcc, and avg F1, respectively, surpassing Non-Adaptation and other TTA methods, underscoring the importance of tabular-specific strategies. Notably, PFT₃A further exceeds FTAT by 4.34%, 1.85%, and 6.27% in these metrics, demonstrating its superior effectiveness in prior-free TTA scenarios. Overall, PFT₃A achieves consistent performance gains across datasets and metrics, validating its effectiveness.

As shown in Table 8, with the MLP backbone, PFT₃A demonstrates strong performance across most datasets. For example, in the Health Insurance dataset, it achieves an accuracy of 73.09% and a balanced accuracy (BAcc) of 73.63%, highlighting its effectiveness in both overall and balanced classification metrics. Similarly, in the ASSIST dataset, PFT₃A achieves an accuracy of 63.54%, showcasing its capability even with a simpler architecture like MLP. When applied with the FT-Transformer backbone, PFT₃A continues to perform robustly, as shown in Table 9. For instance, in the Health Insurance dataset, it achieves an accuracy of 73.88%, further validating its adaptability and effectiveness. The consistent performance improvements across datasets and backbones in Tables 6, 8, and 9 underscore the generalizability of PFT₃A in addressing the challenges of tabular TTA.

## 5.3 EXPERIMENTAL ANALYSIS

**Ablation Study.** We ablate the three components of PFT₃A: Class Prior Estimating (CPE), Robust Feature Learning (RFL), and Representative Subspace Exploration (RSE) to analyze their contributions. The results are in Table 4. Without each component, the performance is

Table 4: Ablation (Acc) with TabTransformer.

| Method | HELOC | ANES | Health Ins. | ASSIST | Hypertension |
|---|---|---|---|---|---|
| PFT₃A w/o CPE | 60.46 | 79.41 | 58.29 | 53.14 | 59.45 |
| PFT₃A w/o RFL | 65.94 | 79.33 | 73.42 | 58.39 | 62.71 |
| PFT₃A w/o RSE | 65.74 | 80.11 | 73.40 | 58.53 | 62.58 |
| PFT₃A | 66.17 | 80.33 | 74.13 | 59.29 | 63.03 |

degraded, indicating that they are essential for the performance of PFT₃A. This highlights the effectiveness of the proposed components in Section 4.1, Section 4.2, and Section 4.3, for estimating class priors, aligning feature distributions, and exploring representative subspaces.

**Varying Parameters.** We analyze the impact of PFT₃A's hyperparameters, including $\beta_1$, $\beta_2$, $\zeta$, and $m$, on performance. $\beta_1$ and $\beta_2$ control the weights of the feature alignment loss and entropy minimization loss in Eq. 16, respectively. $\zeta$ is associated with $\epsilon$, which influences class prior estimation as defined in Eq. 3, while $m$ determines the dimensionality of subspaces in the Representative Subspace Exploration module. The results in Fig. 4 show that performance improves initially and then declines as these parameters vary, highlighting the importance of proper tuning to balance the contributions of the components.

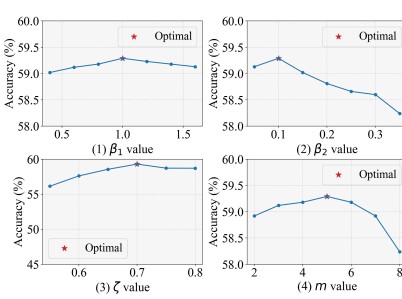

Figure 4: Sensitivity of $\beta_1$, $\beta_2$, $\zeta$, $m$.

## 6    CONCLUSION

In this paper, we developed PFT$_3$A, a novel tabular test-time adaptation (TTA) method that operates without prior knowledge of the source domain. PFT$_3$A is designed to address the challenges of label shift and feature shift in tabular data, which are often exacerbated by the lack of access to source domain data or priors. By incorporating a Class Prior Estimation module, a Robust Feature Learning module, and a Representative Subspace Exploration module, PFT$_3$A effectively mitigates the impact of these shifts, enabling robust and accurate predictions in target domains. Extensive experiments demonstrate the effectiveness of PFT$_3$A across various tabular TTA tasks, showcasing its ability to adapt to dynamic environments without relying on prior knowledge or source data.

## 7    ACKNOWLEDGMENT

This work is supported by grants from the Research Grants Council of Hong Kong Special Administrative Region, China (No. PolyU 15205224), and Smart Cities Research Institute (SCRI) P0051036-P0050643.

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

# A    APPENDIX

## A.1    REPRODUCIBILITY

**Data Availability.**    All real-world datasets used in our experiments are based on Tableshift (Gardner et al., 2023).

## A.2    USE OF LLMS

We only sought assistance from LLMs for language polishing.

## A.3    DETAILED EXPERIMENTAL SETUP

### A.3.1    DETAILED METRICS

Following FTAT (Zhou et al., 2024), we use accuracy (Acc), balanced accuracy (BAcc), and F1 score to evaluate the performance. Accuracy measures the proportion of correctly classified instances out of the total instances, providing a general overview of model performance. However, it may be misleading in imbalanced datasets where one class dominates. Balanced accuracy addresses this limitation by averaging the recall of each class, ensuring a fair evaluation across imbalanced classes. F1 score combines precision and recall into a single metric, offering a balanced measure of a model's ability to correctly identify positive instances while minimizing false positives and false negatives. Together, these metrics provide a comprehensive assessment of model performance across different aspects, particularly in scenarios involving class imbalance or varying misclassification costs.

### A.3.2    DETAILED HYPERPARAMETER SET

In our experiments, we intentionally used the same hyperparameter values across all datasets to demonstrate the generalizability and robustness of PFT$_3$A. Specifically, we maintained identical hyperparameters ($\beta_1$=1.0, $\beta_2$=0.1, $m$=5, $\zeta$=0.7) for all five tabular datasets in the Tableshift benchmark. The entropy threshold $\epsilon$ plays a critical role in distinguishing between source-like and target-like data samples. Rather than using a fixed threshold value across all datasets, we employed a data-adaptive approach. For each incoming batch of target data $D_t^j$, we calculated the entropy distribution of model predictions and set threshold as a specific percentile of this distribution. Through empirical validation, we found that the 70th percentile ($\zeta$=0.7) consistently provided a balanced separation between confident and uncertain predictions across various tabular datasets.

### A.3.3    DETAIL DATASETS

Based on Tableshift (Gardner et al., 2023), a summary of datasets we use is shown in Table 5. We give a detailed overview of each dataset, including task target, shift source, Acc Gap, feature shift level,

and label shift level. Acc Gap $\Delta_{Acc}$denotes accuracy difference between test set with distribution shift $D_t^{OOD}$ and without distribution shift $D_t^{ID}$, which is defined by

$$\Delta_{Acc} = Acc(f_{\theta_0}, D_t^{ID}) - Acc(f_{\theta_0}, D_t^{OOD}),  \tag{17}$$

where $f_{\theta_0}$ denotes the model trained on source data $D_s$. $\Delta_{Acc}$ incurred when training a classifier is comprised of changes in $p(x)$ ("feature shift") and changes in $p(y)$ ("label shift"). The feature shift level is measured with Optimal Transport Dataset Distance with the Gaussian approximation Alvarez-Melis & Fusi (2020). The label shift level is measured based on the label distribution of source and target data.

Table 5: Summary of datasets and their associated distribution shifts.

| Dataset | Task Target | Shift Source | Acc Gap | feature shift | label shift |
|---|---|---|---|---|---|
| ASSIST | Next Answer Correct | School | -34.49% | 24,054.59 | 0.0670 |
| ANES | VotedinU.S.presidential election | Geographic Region | -2.58% | 13.60 | 0.0025 |
| HELOC | Repayment of Home Equity Line of Credit loan | Est. third-party risk level | -22.58% | 19.35 | 0.0983 |
| Hypertension | Hypertension diagnosis for high-risk age(50+) | BMI Category | -4.36% | 4.69 | 0.0022 |
| Health Ins. | Coverage of non-Medicare eligible low-income individuals | Disability Status | -14.46% | 5.79 | 0.1701 |

**HELOC.** The Home Equity Line of Credit (HELOC) is a part of credit secured by the applicant's home. A HELOC provides access to a revolving credit line with a lower interest rate than other types of loans. To assess an applicant's suitability for HELOC, a lender evaluates an applicants' financial background to predict whether a given applicant islikely to repay a line of credit and, if so, how much credit should be extended. In order to accurate credit risk predictions for their overall utility for both lenders and borrowers and achieve equal treatment, HELOC dataset which contains 10,459 samples need to predict whether a consumer is 90 days past due or worse at least once over a period of 24 months using 38 features varying from financial activity to credit inquiries.

**ANES.** Understanding participation in elections is a critical task for policymakers, politicians, and those with an interest in democracy. Predicting which individuals will vote in an electio, is widely acknowledged as critical to polling and campaigning in U.S. politics. For better understanding individual activity in presidential election, ANES dataset predict whether an individual will vote in the U.S presidential election using 365 features including voting behavior, elections, public opinion and attitudes. The ANES dataset includes 60,377 samples.

**ASSIST.** ASSISTMENTS dataset in education field (abbreviated as ASSIST). The ASSISTments tutoring platform is a free, web-based, data-driven tutoring platform for students in grades 3-12. ASSISTMENTS dataset contains affect predictions such as such as boredom, confusion, frustration, and engaged problem-solving behavior on students who use the ASSISTMENTS tutoring platform. The numbers of features and samples in ASSISTMENTS dataset are 26 and 2,667,776.

**Hypertension.** Hypertension, or systolic blood pressure (typically systolic pressure 130 mm Hg or higher or diastolic 80 or higher) affects nearly half of Americans. When left untreated, hypertension is associated with the strongest evidence for causation of all risk factors for heart attack and other cardiovascular disease (Fuchs and Whelton 2020). As a result, it is important to predict blood pressure accurately and efficitively. Hypertension dataset has a goal to achieve efficitive blood pressure measurement and increase the prediction accuracy. Hypertension dataset contains 846,781 samples with 100 features related to several risk factors for hypertension.

**Health Ins.** Public health insurance makes a significant performance in providing affordable and accessible medical care for individuals. A high level of health insurance ownership is important for the healthy development of the individual. So, it is important to raise the rate of owning health insurance. Health Ins. dataset is related to public coverage field and the goal is to predict whether an individual is covered by public health insurance using 135 features. The number of samples in Health Ins. dataset is 5,916,565.

Table 6: Performance comparison with TabTransformer as backbone. The best is in bold.

| Method | HELOC | | | ANES | | | Health Ins. | | |
|---|---|---|---|---|---|---|---|---|---|
| | Acc. | BAcc. | F1 | Acc. | BAcc. | F1 | Acc. | BAcc. | F1 |
| Non-Adaptation | $55.66 \pm 1.34$ | $59.60 \pm 1.11$ | $44.30 \pm 3.08$ | $78.95 \pm 0.27$ | $75.30 \pm 0.45$ | $84.23 \pm 0.12$ | $65.35 \pm 1.08$ | $70.11 \pm 0.67$ | $65.88 \pm 1.63$ |
| TENT | $56.37 \pm 1.55$ | $59.60 \pm 1.11$ | $50.34 \pm 5.91$ | $78.82 \pm 0.38$ | $75.18 \pm 0.56$ | $84.14 \pm 0.19$ | $65.22 \pm 1.18$ | $70.07 \pm 0.70$ | $65.62 \pm 1.82$ |
| EATA | $56.37 \pm 1.55$ | $57.04 \pm 4.04$ | $52.57 \pm 4.42$ | $78.82 \pm 0.38$ | $75.18 \pm 0.56$ | $84.14 \pm 0.19$ | $65.35 \pm 1.08$ | $70.11 \pm 0.67$ | $65.88 \pm 1.63$ |
| LAME | $51.71 \pm 4.68$ | $59.56 \pm 1.51$ | $44.30 \pm 6.64$ | $78.57 \pm 0.49$ | $74.77 \pm 0.80$ | $84.04 \pm 0.20$ | $65.35 \pm 1.08$ | $70.11 \pm 0.67$ | $65.88 \pm 1.63$ |
| CoTTA | $56.37 \pm 1.55$ | $59.60 \pm 1.11$ | $50.34 \pm 5.91$ | $78.82 \pm 0.38$ | $75.18 \pm 0.56$ | $84.14 \pm 0.19$ | $65.35 \pm 1.08$ | $70.11 \pm 0.67$ | $65.88 \pm 1.63$ |
| ODS | $52.19 \pm 4.15$ | $56.77 \pm 3.56$ | $44.70 \pm 6.56$ | $78.48 \pm 0.70$ | $74.69 \pm 1.09$ | $83.97 \pm 0.31$ | $57.14 \pm 5.22$ | $64.75 \pm 3.23$ | $51.33 \pm 10.03$ |
| SAR | $56.37 \pm 1.55$ | $59.60 \pm 1.11$ | $50.34 \pm 5.91$ | $78.82 \pm 0.38$ | $75.18 \pm 0.56$ | $84.14 \pm 0.19$ | $65.35 \pm 1.08$ | $70.11 \pm 0.67$ | $65.88 \pm 1.63$ |
| FTAT | $60.54 \pm 0.10$ | $63.22 \pm 0.09$ | $55.81 \pm 0.15$ | $79.46 \pm 0.03$ | $76.29 \pm 0.05$ | $\mathbf{84.34 \pm 0.02}$ | $66.31 \pm 0.01$ | $70.84 \pm 0.02$ | $67.19 \pm 0.01$ |
| PFT$_3$A (ours) | $\mathbf{66.17 \pm 0.17}$ | $\mathbf{65.17 \pm 0.27}$ | $\mathbf{70.91 \pm 0.78}$ | $\mathbf{80.33 \pm 0.15}$ | $\mathbf{78.72 \pm 0.17}$ | $84.06 \pm 0.33$ | $\mathbf{74.13 \pm 0.78}$ | $\mathbf{72.67 \pm 0.36}$ | $\mathbf{79.33 \pm 1.03}$ |

| Method | ASSIST | | | Hypertension | | | Avg | | |
|---|---|---|---|---|---|---|---|---|---|
| | Acc. | BAcc. | F1 | Acc. | BAcc. | F1 | Acc. | BAcc. | F1 |
| Non-Adaptation | $49.04 \pm 4.23$ | $53.07 \pm 2.12$ | $58.92 \pm 2.34$ | $54.87 \pm 0.89$ | $59.24 \pm 0.63$ | $46.25 \pm 2.16$ | $60.77 \pm 1.56$ | $63.46 \pm 1.00$ | $59.92 \pm 1.87$ |
| TENT | $47.83 \pm 3.39$ | $53.11 \pm 2.18$ | $61.27 \pm 0.32$ | $41.74 \pm 0.01$ | $50.12 \pm 0.01$ | $0.78 \pm 0.06$ | $58.00 \pm 1.30$ | $61.62 \pm 0.91$ | $52.43 \pm 1.66$ |
| EATA | $45.28 \pm 0.16$ | $51.44 \pm 0.14$ | $61.40 \pm 0.41$ | $54.86 \pm 0.89$ | $59.24 \pm 0.63$ | $46.24 \pm 2.15$ | $59.94 \pm 0.81$ | $62.60 \pm 1.21$ | $62.05 \pm 1.76$ |
| LAME | $45.12 \pm 3.46$ | $51.30 \pm 2.20$ | $61.40 \pm 0.41$ | $54.87 \pm 0.89$ | $59.24 \pm 0.63$ | $46.25 \pm 2.16$ | $59.12 \pm 2.12$ | $63.00 \pm 1.16$ | $60.37 \pm 2.21$ |
| CoTTA | $45.51 \pm 0.39$ | $51.64 \pm 0.34$ | $\mathbf{61.56 \pm 0.16}$ | $54.87 \pm 0.89$ | $59.24 \pm 0.63$ | $46.25 \pm 2.16$ | $60.18 \pm 0.86$ | $63.15 \pm 0.66$ | $61.63 \pm 2.01$ |
| ODS | $45.12 \pm 3.46$ | $51.30 \pm 2.20$ | $61.40 \pm 0.41$ | $54.87 \pm 0.89$ | $59.24 \pm 0.63$ | $46.25 \pm 2.16$ | $57.56 \pm 2.88$ | $61.35 \pm 2.14$ | $57.53 \pm 3.89$ |
| SAR | $45.12 \pm 3.46$ | $51.30 \pm 2.20$ | $61.40 \pm 0.41$ | $54.87 \pm 0.89$ | $59.24 \pm 0.63$ | $46.25 \pm 2.16$ | $60.11 \pm 1.47$ | $63.09 \pm 1.03$ | $61.60 \pm 2.06$ |
| FTAT | $53.17 \pm 2.05$ | $55.85 \pm 2.02$ | $58.94 \pm 1.62$ | $61.78 \pm 0.02$ | $61.65 \pm 0.05$ | $65.62 \pm 0.03$ | $64.25 \pm 0.44$ | $65.57 \pm 0.45$ | $66.38 \pm 0.37$ |
| PFT$_3$A (ours) | $\mathbf{59.29 \pm 0.31}$ | $\mathbf{59.15 \pm 0.39}$ | $59.73 \pm 1.13$ | $\mathbf{63.03 \pm 0.11}$ | $\mathbf{61.39 \pm 0.08}$ | $\mathbf{69.20 \pm 0.37}$ | $\mathbf{68.59 \pm 0.30}$ | $\mathbf{67.42 \pm 0.25}$ | $\mathbf{72.65 \pm 0.73}$ |

### A.3.4 DETAIL COMPARED METHODS

We compare our PFT$_3$A with various TTA methods, including typical TTA methods (i.e., TENT (Wang et al., 2020) and EATA (Niu et al., 2022)), continual TTA methods (i.e., CoTTA (Wang et al., 2022)), robust FTTA methods (i.e., SAR (Niu et al., 2023), LAME (Boudiaf et al., 2022), ODS (Zhou et al., 2023)), and tabular TTA method (i.e., FTAT (Zhou et al., 2024)).

- **TENT** (Wang et al., 2020) updates the model parameters with entropy minimization loss.
- **EATA** (Niu et al., 2022) combines active sample selection with Fisher regularization to enhance prediction performance while mitigating catastrophic forgetting.
- **CoTTA** (Wang et al., 2022) mitigates error accumulation in streaming data by leveraging weight-and-augmentation averaged pseudo-labels and stochastic parameter restoration.
- **SAR** (Niu et al., 2023) filters samples based on test entropy and optimizes model parameters toward flat minima to enhance robustness and performance.
- **LAME** (Boudiaf et al., 2022) adjusts model predictions through a cautious adaptation strategy.
- **ODS** (Zhou et al., 2023) disentangles the combined distribution shift, tackling covariate and label shifts independently.
- **FTAT** (Zhou et al., 2024) optimizes label distributions, filters low-quality predictions, and employs online ensemble learning for robust predictions.

## A.4 DETAILED EXPERIMENTS AND ANALYSIS

### A.4.1 DETAILED EXPERIMENTAL RESULTS

Table 6, Table 8, and Table 9 show the mean and standard deviation with TabTransformer (Huang et al., 2020), MLP, and FT-Transformer (Gorishniy et al., 2021) by three repeated experiments. The results show that our PFT$_3$A achieves the best performance on majority of datasets and metrics, demonstrating the effectiveness of our approach.

Table 7: Cost analysis on PFT$_3$A and FTAT.

| Metric | PFT$_3$A (Ours) | FTAT (Baseline) | Difference | Relative Improvement |
|---|---|---|---|---|
| Accuracy (ACC) | 59.29% | 53.17% | +6.12% | 11.5% |
| Runtime (s) | 1.0161 | 0.9346 | +0.0815 | 8.7% |
| Memory (MB) | 1074.62 | 1050.46 | +24.16 | 2.3% |

Table 8: Performance comparison with MLP as backbone. The best is in bold.

| Method | HELOC | | | ANES | | | Health Ins. | | |
|---|---|---|---|---|---|---|---|---|---|
| | Acc. | BAcc. | F1 | Acc. | BAcc. | F1 | Acc. | BAcc. | F1 |
| Non-Adaptation | 54.37 ± 5.35 | 58.25 ± 3.56 | 40.02 ± 16.8 | 79.11 ± 0.31 | 75.66 ± 0.46 | 84.24 ± 0.16 | 65.79 ± 0.63 | 70.68 ± 0.44 | 66.21 ± 0.90 |
| TENT | 54.35 ± 5.38 | 58.24 ± 3.58 | 39.95 ± 16.9 | 78.07 ± 0.35 | 74.09 ± 0.65 | 83.76 ± 0.13 | 64.30 ± 0.70 | 69.79 ± 0.47 | 63.87 ± 1.06 |
| EATA | 54.37 ± 5.35 | 58.25 ± 3.56 | 40.02 ± 16.8 | 78.13 ± 0.30 | 74.20 ± 0.59 | 83.79 ± 0.10 | 65.78 ± 0.63 | 70.68 ± 0.44 | 66.21 ± 0.90 |
| LAME | 43.10 ± 0.00 | 50.00 ± 0.00 | 30.10 ± 0.00 | 63.50 ± 0.00 | 54.60 ± 0.00 | 46.80 ± 0.00 | 63.44 ± 1.69 | 69.14 ± 1.09 | 62.61 ± 2.69 |
| CoTTA | 54.36 ± 5.35 | 58.25 ± 3.56 | 40.03 ± 16.8 | 78.13 ± 0.30 | 74.20 ± 0.59 | 83.79 ± 0.10 | 65.79 ± 0.63 | 70.68 ± 0.44 | 66.21 ± 0.90 |
| ODS | 43.10 ± 0.00 | 50.00 ± 0.00 | 30.10 ± 0.00 | 63.50 ± 0.00 | 54.60 ± 0.00 | 46.80 ± 0.00 | 63.45 ± 1.68 | 69.14 ± 1.07 | 62.62 ± 2.68 |
| SAR | 52.32 ± 6.05 | 56.74 ± 3.99 | 33.16 ± 19.0 | 78.13 ± 0.30 | 74.20 ± 0.59 | 83.79 ± 0.10 | 65.79 ± 0.63 | 70.68 ± 0.44 | 66.21 ± 0.90 |
| FTAT | 60.89 ± 0.15 | 62.67 ± 0.15 | 59.14 ± 0.13 | 79.29 ± 0.11 | 75.83 ± 0.19 | **84.42 ± 0.10** | 67.25 ± 0.13 | 71.60 ± 0.13 | 68.38 ± 0.11 |
| PFT$_3$A (ours) | **65.26 ± 0.09** | **62.87 ± 0.25** | **72.43 ± 0.16** | **79.84 ± 0.09** | **79.23 ± 0.02** | 82.94 ± 0.15 | **73.09 ± 0.08** | **73.63 ± 0.01** | **77.20 ± 0.12** |

| Method | ASSIST | | | Hypertension | | | Avg | | |
|---|---|---|---|---|---|---|---|---|---|
| | Acc. | BAcc. | F1 | Acc. | BAcc. | F1 | Acc. | BAcc. | F1 |
| Non-Adaptation | 55.86 ± 3.81 | 60.81 ± 3.37 | 66.42 ± 1.86 | 58.76 ± 1.68 | 61.69 ± 0.95 | 55.46 ± 4.03 | 62.78 ± 2.36 | 65.42 ± 1.76 | 62.47 ± 4.75 |
| TENT | 50.87 ± 0.32 | 56.41 ± 0.29 | 63.99 ± 0.15 | 41.67 ± 0.08 | 50.07 ± 0.05 | 0.49 ± 0.36 | 57.85 ± 1.37 | 61.72 ± 1.01 | 50.41 ± 3.72 |
| EATA | 55.86 ± 0.18 | 60.81 ± 0.16 | 66.42 ± 0.08 | 57.81 ± 2.32 | 61.19 ± 1.38 | 52.87 ± 5.82 | 62.39 ± 1.76 | 65.03 ± 1.23 | 61.86 ± 4.74 |
| LAME | 45.12 ± 0.18 | 51.30 ± 0.18 | 61.40 ± 0.18 | 58.63 ± 1.60 | 61.64 ± 0.92 | 55.12 ± 3.84 | 54.76 ± 0.69 | 57.34 ± 0.44 | 51.21 ± 1.34 |
| CoTTA | 55.86 ± 0.18 | 60.81 ± 0.16 | 66.42 ± 0.08 | 58.76 ± 1.68 | 61.69 ± 0.95 | 55.46 ± 4.03 | 62.58 ± 1.63 | 65.13 ± 1.14 | 62.38 ± 4.38 |
| ODS | 45.12 ± 0.18 | 51.30 ± 0.18 | 61.40 ± 0.18 | 57.12 ± 1.46 | 60.80 ± 0.93 | 51.41 ± 3.43 | 54.46 ± 0.66 | 57.17 ± 0.44 | 50.47 ± 1.26 |
| SAR | 55.86 ± 0.18 | 60.81 ± 0.16 | 66.42 ± 0.08 | 58.21 ± 1.51 | 61.50 ± 0.77 | 53.81 ± 4.05 | 62.06 ± 1.73 | 64.79 ± 1.19 | 60.68 ± 4.83 |
| FTAT | 51.84 ± 2.80 | 57.26 ± 1.18 | 64.45 ± 0.80 | 63.28 ± 0.54 | **63.18 ± 0.52** | 66.99 ± 1.10 | 64.51 ± 0.75 | 66.11 ± 0.43 | 68.68 ± 0.45 |
| PFT$_3$A (ours) | **63.54 ± 0.90** | **65.57 ± 1.22** | **66.45 ± 0.94** | **64.53 ± 0.22** | 61.83 ± 0.35 | **71.93 ± 1.22** | **69.25 ± 0.11** | **68.63 ± 0.37** | **74.19 ± 0.52** |

Table 9: Performance comparison with FT-Transformer as backbone. The best is in bold.

| Method | HELOC | | | ANES | | | Health Ins. | | |
|---|---|---|---|---|---|---|---|---|---|
| | Acc. | BAcc. | F1 | Acc. | BAcc. | F1 | Acc. | BAcc. | F1 |
| Non-Adaptation | 46.26 ± 1.05 | 52.48 ± 0.78 | 13.32 ± 4.50 | 75.47 ± 1.31 | 71.50 ± 2.06 | **81.80 ± 0.45** | 58.33 ± 4.05 | 65.44 ± 2.56 | 54.06 ± 7.72 |
| TENT | 44.98 ± 1.83 | 51.45 ± 1.39 | 8.11 ± 7.80 | 63.02 ± 0.89 | 54.52 ± 1.15 | 76.19 ± 0.40 | 36.44 ± 0.03 | 50.05 ± 0.02 | 0.24 ± 0.11 |
| EATA | 45.95 ± 1.00 | 52.23 ± 0.74 | 12.27 ± 4.38 | 74.65 ± 1.66 | 70.16 ± 2.49 | 81.51 ± 0.67 | 57.40 ± 4.35 | 64.86 ± 2.79 | 52.23 ± 8.49 |
| LAME | 43.14 ± 0.04 | 50.03 ± 0.03 | 0.20 ± 0.14 | 75.37 ± 1.34 | 71.35 ± 2.12 | 81.73 ± 0.48 | 59.08 ± 4.57 | 65.44 ± 2.60 | 55.91 ± 9.04 |
| CoTTA | 46.26 ± 1.05 | 52.48 ± 0.78 | 10.67 ± 7.65 | 75.47 ± 1.31 | 71.50 ± 2.06 | **81.80 ± 0.45** | 58.33 ± 4.05 | 65.44 ± 2.56 | 54.06 ± 7.72 |
| ODS | 43.14 ± 0.04 | 50.03 ± 0.03 | 0.20 ± 0.14 | 75.41 ± 1.39 | 71.41 ± 2.15 | 81.75 ± 0.54 | 59.99 ± 3.45 | 65.54 ± 1.53 | 58.37 ± 6.92 |
| SAR | 43.30 ± 0.00 | 50.20 ± 0.00 | 30.60 ± 0.00 | 75.47 ± 1.31 | 71.50 ± 2.06 | **81.80 ± 0.45** | 58.33 ± 4.05 | 65.44 ± 2.56 | 54.06 ± 7.72 |
| FTAT | 59.20 ± 0.16 | 61.51 ± 0.13 | 55.52 ± 0.34 | 76.06 ± 0.21 | 73.29 ± 0.17 | 81.38 ± 0.23 | 66.45 ± 0.03 | 70.50 ± 0.02 | 67.85 ± 0.04 |
| PFT$_3$A (ours) | **65.46 ± 0.12** | **62.70 ± 0.25** | **73.17 ± 0.20** | **76.62 ± 0.17** | **75.42 ± 0.45** | 80.60 ± 0.66 | **73.88 ± 0.67** | **72.84 ± 0.19** | **78.87 ± 1.05** |

| Method | ASSIST | | | Hypertension | | | Avg | | |
|---|---|---|---|---|---|---|---|---|---|
| | Acc. | BAcc. | F1 | Acc. | BAcc. | F1 | Acc. | BAcc. | F1 |
| Non-Adaptation | 58.32 ± 0.07 | 62.99 ± 0.09 | **67.63 ± 0.09** | 58.88 ± 0.28 | 61.84 ± 0.14 | 55.71 ± 0.79 | 59.45 ± 1.35 | 62.85 ± 1.13 | 54.50 ± 2.71 |
| TENT | 58.25 ± 0.07 | 62.91 ± 0.09 | 67.57 ± 0.09 | 47.01 ± 6.39 | 53.88 ± 4.52 | 18.83 ± 20.89 | 54.05 ± 1.84 | 57.84 ± 1.43 | 48.68 ± 5.86 |
| EATA | 48.04 ± 0.25 | 53.85 ± 0.19 | 62.60 ± 0.07 | 58.84 ± 0.27 | 61.82 ± 0.13 | 55.62 ± 0.78 | 57.23 ± 1.51 | 60.24 ± 1.27 | 52.85 ± 2.88 |
| LAME | 56.54 ± 2.48 | 62.98 ± 0.06 | **67.63 ± 0.07** | 58.78 ± 0.27 | 61.78 ± 0.15 | 55.47 ± 0.74 | 59.40 ± 1.74 | 62.06 ± 0.99 | 51.16 ± 2.09 |
| CoTTA | 58.25 ± 0.07 | 62.91 ± 0.09 | 67.57 ± 0.09 | 58.88 ± 0.28 | 61.84 ± 0.14 | 55.71 ± 0.79 | 59.44 ± 1.35 | 62.83 ± 1.13 | 53.96 ± 3.34 |
| ODS | 57.39 ± 1.19 | 62.16 ± 1.04 | 67.14 ± 0.58 | 58.77 ± 0.26 | 61.78 ± 0.14 | 55.45 ± 0.72 | 58.94 ± 1.27 | 62.18 ± 0.98 | 52.58 ± 1.78 |
| SAR | 58.25 ± 0.07 | 62.91 ± 0.09 | 67.57 ± 0.09 | 59.64 ± 0.65 | 62.24 ± 0.29 | 57.52 ± 1.63 | 59.00 ± 1.22 | 62.46 ± 1.00 | 58.31 ± 1.98 |
| FTAT | 58.32 ± 0.05 | 62.95 ± 0.05 | 67.56 ± 0.03 | 62.34 ± 0.07 | 63.30 ± 0.08 | 64.13 ± 0.06 | 64.47 ± 0.10 | 66.31 ± 0.09 | 67.29 ± 0.14 |
| PFT$_3$A (ours) | **59.36 ± 0.15** | **63.53 ± 0.11** | 67.43 ± 0.22 | **64.74 ± 0.23** | **63.07 ± 0.38** | **70.74 ± 1.00** | **68.01 ± 0.27** | **67.51 ± 0.28** | **74.16 ± 0.63** |

### A.4.2 DETAILED COST ANALYSIS

To ensure a fair and reproducible comparison about cost, we evaluated both our method (PFT$_3$A) and the baseline method (FTAT) on ASSIST with TabTransformer. The quantitative comparison in Table 7 confirms that PFT$_3$A achieves significantly improved accuracy with minimal additional resource consumption. This balance makes our method particularly suitable for practical applications where prediction quality is paramount. The results strengthen our claims regarding the method's practical utility and deployment readiness.

### A.4.3 DETAILED ABLATION ANALYSIS

**Class Prior Estimating (CPE).** When CPE is disabled (PFT$_3$A w/o CPE), performance sharply deteriorates in Health Insurance (Acc drops from 74.13% to 58.29%) and Hypertension (Acc decreases from 63.03% to 59.45%), indicating that direct adaptation without class prior estimation amplifies prediction errors. Notably, in Health Insurance, the absence of CPE even underperforms the Non-Adaptation baseline (58.29% vs. 65.35%), suggesting that naive feature alignment without class prior estimation still shows poor performance and Health Insurance exhibits a larger label shift compared with other datasets.

**Robust Feature Learning (RFL).** Robust Feature Learning (RFL) proves essential for handling covariate shift in tabular data. By aligning feature distributions between source and target domains, RFL enables effective adaptation to unseen data. The results clearly demonstrate its impact: on

Table 10: Ablation (Acc) with TabTransformer.

| Method | HELOC | ANES | Health Ins. | ASSIST | Hypertension |
|---|---|---|---|---|---|
| Non-Adaptation | 55.66 | 78.95 | 65.35 | 49.04 | 54.87 |
| PFT$_3$A w/o CPE | 60.46 | 79.41 | 58.29 | 53.14 | 59.45 |
| PFT$_3$A w/o RFL | 65.94 | 79.33 | 73.42 | 58.39 | 62.71 |
| PFT$_3$A w/o RSE | 65.74 | 80.11 | 73.40 | 58.53 | 62.58 |
| PFT$_3$A | 66.17 | 80.33 | 74.13 | 59.29 | 63.03 |

ANES, accuracy improves from 79.33% (w/o RFL) to 80.33% (with RFL), while ASSISTMENTS shows gains from 58.39% to 59.29%. While the performance gains from RFL appear less pronounced, this primarily reflects the Tableshift benchmark's characteristic where label shift dominates feature shift. Nevertheless, consistent improvements of RFL across datasets confirm RFL's critical role in mitigating feature shifts. Moreover, RFL helps subsequent subspace exploration by providing well-aligned feature representations.

**Representative Subspace Exploration (RSE).** The ablation results in Table 10 demonstrate RSE's indispensable role in refining feature alignment for tabular TTA. Disabling RSE (PFT$_3$A w/o RSE) leads to consistent performance drops across all datasets, most notably in Health Insurance (Acc: 74.13% to 73.40%) and Hypertension (Acc: 63.03% to 62.58%). These results reflect RSE's unique ability to eliminate redundant features and enhance feature alignment.

### A.4.4 HYPERPARAMETER SENSITIVITY AND SELECTION

We determine a single default hyperparameter configuration for PFT$_3$A via grid search on the ASSIST dataset using a TabTransformer backbone. The search space is defined as: $\beta_1 \in \{0.2, 0.4, 0.6, 0.8, 1.0, 1.2, 1.4, 1.6, 1.8\}$, $\beta_2 \in \{0.05, 0.1, 0.15, 0.2, 0.25, 0.3, 0.5\}$, $\zeta \in \{0.55, 0.6, 0.65, 0.7, 0.75, 0.8, 0.85\}$, and $m \in \{2, 3, 4, 5, 6, 7, 8\}$. The configuration achieving the best average target-domain performance on ASSIST is selected as the default: $\beta_1 = 1.0, \beta_2 = 0.10, \zeta = 0.70, m = 5$. Once selected, this configuration is fixed and directly applied to all other datasets (HELOC, ANES, Health Insurance, and Hypertension) without any dataset-specific tuning. This protocol strictly follows the realistic TTA setting, where target labels are unavailable and hyperparameter adaptation per dataset is infeasible. Figure 4 shows that while extreme values can degrade performance, PFT$_3$A exhibits a relatively stable performance plateau around the selected default configuration. The selected hyperparameters lie within this stable region, indicating that the method is not overly sensitive to moderate variations and does not require fine-grained tuning.

To verify that PFT$_3$A does not rely on dataset-specific tuning, we compare the performance of the fixed ASSIST-tuned configuration with dataset-specific optimal configurations obtained using the same search space on each dataset, as shown in Table 11.

Table 11: Comparison between the fixed ASSIST-tuned configuration and dataset-specific optimal hyperparameters.

| Dataset | FTAT | Fixed Config | Dataset-Specific Best | Gap |
|---|---|---|---|---|
| HELOC | 60.54 | 66.17 | 66.27 | 0.10 |
| ANES | 79.46 | 80.33 | 80.42 | 0.09 |
| Health Ins. | 66.31 | 74.13 | 74.34 | 0.21 |
| Hypertension | 61.78 | 63.03 | 63.09 | 0.06 |

According to Table 11, the performance gap between the fixed configuration and dataset-specific optimal settings remains small, demonstrating that PFT$_3$A generalizes well across domains without requiring per-dataset hyperparameter tuning. These findings confirm the practical applicability of PFT$_3$A in realistic test-time adaptation scenarios. Furthermore, regardless of whether the fixed configuration or dataset-specific optimal settings are used, the performance of PFT$_3$A consistently outperforms the current state-of-the-art baseline, FTAT, by a significant margin.

Table 12: Comparison of our estimated source domain class priors with the ground truth.

|              | HELOC        | ANES         | Health Ins.  | ASSIST       | Hypertension |
|--------------|--------------|--------------|--------------|--------------|--------------|
| Ground Truth | (0.74, 0.26) | (0.31, 0.69) | (0.78, 0.22) | (0.31, 0.69) | (0.60, 0.40) |
| Estimation   | (0.89, 0.11) | (0.31, 0.69) | (0.63, 0.37) | (0.29, 0.71) | (0.66, 0.34) |
| Error        | 0.15         | 0.00         | 0.15         | 0.02         | 0.06         |

### A.4.5 CLASS PRIOR ESTIMATION ANALYSIS

Table 12 compares the true source class priors with the class priors estimated by our CPE module. The estimation errors of our CPE module across the five datasets are 0.15, 0.00, 0.15, 0.02, and 0.06, respectively, demonstrating its reliability in estimating class priors.

### A.5 LIMITATIONS AND FUTURE WORK

When the source model is manipulated by adversarial attacks, the performance of $PFT_3A$ may be compromised since the target model relies on the source model. As a future work, we plan to investigate the robustness of $PFT_3A$ against adversarial attacks and explore methods to enhance its resilience in such scenarios. In cases of severe distributional shifts, $PFT_3A$ may experience performance degradation. Dynamic thresholds that adjust based on shift severity detection could improve robustness. In applications where test data arrives in very small batches, $PFT_3A$ may face several challenges. Maintaining a sliding window of recent batches could provide more stable statistical estimates.

