# OpenReview forum: "Prior-free Tabular Test-time Adaptation"
_ICLR.cc/2026/Conference — ICLR 2026 Poster_

### Official Review · Reviewer_B7xd · 2025-10-16

**Soundness:** 3
**Presentation:** 4
**Contribution:** 3
**Rating:** 8
**Confidence:** 4

**Summary:**

This paper focuses on the tabular TTA problem without the accessibility of the source domain or prior. Based on the analysis, this paper proposes PFT3A method, which consists of a Class Prior Estimating module for estimating source-target class priors, Robust Feature
Learning module for learning robust features by aligning source-like and target-like features to mitigate feature shift, and Representative Subspace Exploration module for selecting informative features. Experiments conducted in this paper show the effectiveness of PFT3A under prior-free tabular test-time adaptation problem.

**Strengths:**

1. This paper is well-written, which makes it easy to follow. The motivation is clear, and the math symbol is easy to follow.
2. The method introduced in this paper is novel; The idea of optimizing the Kullback-Leibler (KL) divergence of the source distribution and the target distribution with informative columns provides insights for tabular machine learning.
3. The experiment conducted in this paper is well-designed, showing the effectiveness in performance and robustness in parameters of the PFT3A method.

**Weaknesses:**

1. The motivation behind the method’s design needs to be explained. In Eq. 6, the estimation of the prior leverages both the model’s predicted outputs and the covariance matrix of the features. This design links the predicted probabilities with the feature information of the data; however, the paper does not clarify the rationale for combining these two components.
2. Typos:

a. In line 298, the cite format would be better using \citep:
>  Assuming the feature distributions follow a Gaussian distribution Adachi et al.
(2024), --> Assuming the feature distributions follow a Gaussian distribution (Adachi et al. 2024).

b. The symbol $\zeta$ does not appear until section 5.3, and it is not clearly defined. According to the explanation in line 466, does it refer to $\epsilon$, or what is the relationship between $\zeta$ and $\epsilon$?

**Questions:**

See Weakness above.

---

> ### Author Response · Authors · 2025-12-02
> **Response to Reviewer B7xd**
>
> > W1. The motivation behind the method’s design needs to be explained. In Eq. 6, the estimation of the prior leverages both the model’s predicted outputs and the covariance matrix of the features. This design links the predicted probabilities with the feature information of the data; however, the paper does not clarify the rationale for combining these two components.
>
> **Response:**  The integration of predicted probabilities and feature covariance matrices in Equation 6 is grounded in the complementary nature of these two information sources for robust domain adaptation under prior-free conditions. This design addresses a fundamental challenge in test-time adaptation: how to leverage both the model's current belief state (through predictions) and the underlying data structure (through feature representations) to achieve accurate class prior estimation without source domain access.
>
> The predicted probabilities component captures the model's current confidence about sample classifications, which reflects the immediate evidence of label distribution shifts. This is particularly important for detecting and adapting to changes in class proportions between source and target domains. However, relying solely on prediction probabilities can be problematic in cases of severe feature shift or when the model's initial confidence is miscalibrated. The feature covariance matrix component provides crucial information about the underlying data geometry and feature relationships that remain more stable across domains than the raw predictions.
>
> > W2. Typos: a. In line 298, the cite format would be better using \citep. b. The symbol $\zeta$ does not appear until section 5.3, and it is not clearly defined. According to the explanation in line 466, does it refer to $\epsilon$, or what is the relationship between $\zeta$ and $\epsilon$?
>
> **Response:** Thank you for your careful reading and valuable feedback regarding the typos and symbol definitions in our manuscript. In the revised manuscript, we will change the citation to use the appropriate \citep command as suggested, ensuring consistency with the required citation style throughout the paper.
>
> $\epsilon$ represents the entropy threshold​ used to distinguish between source-like and target-like data samples during test-time adaptation. $\zeta$ is the percentile value​ used to determine the entropy threshold $\epsilon$ in a data-adaptive manner. Rather than fixing $\epsilon$ across all datasets, we set it dynamically based on the entropy distribution of each incoming test batch.
>
> By using $\zeta$ to compute $\epsilon$, our method achieves dataset-agnostic adaptation, as the threshold self-tunes to the entropy distribution of any given batch, mitigating biases from pre-defined values.

---

### Official Review · Reviewer_vak8 · 2025-10-27

**Soundness:** 3
**Presentation:** 3
**Contribution:** 2
**Rating:** 4
**Confidence:** 2

**Summary:**

This paper tackles test-time adaptation (TTA) for tabular data in a realistic, prior-free setting where no source data or distributional priors are available. The proposed method, PFT₃A, comprises three key modules: (1) a Class Prior Estimation module that addresses label shift via unsupervised entropy-based techniques; (2) a Robust Feature Learning module that aligns features using KL divergence without relying on confidence thresholding; and (3) a Representative Subspace Exploration module that identifies and adapts discriminative subspaces through PCA-inspired methods. Comprehensive experiments on five datasets with various backbones demonstrate consistent superiority over existing TTA baselines, including prior-free and tabular-specific ones. Ablation studies and sensitivity analyses validate the efficacy of each component.

**Strengths:**

1). The paper identifies and formalizes an under-explored challenge in tabular TTA: the fully prior-free and source-free scenario. This is motivated by empirical evidence in Figure 2, highlighting limitations of prior methods under real-world constraints.

2). PFT₃A integrates unsupervised class prior estimation from prediction entropy (to handle label shift), KL-based feature alignment (for covariate shift), and a principled subspace selection strategy via PCA, which mitigates overfitting to non-discriminative features.

3). The method is evaluated on five TableShift benchmark datasets using three deep tabular backbones, showing improvements over a range of baselines. The inclusion of ablations and analyses enhances the credibility of the results.

**Weaknesses:**

1). Although the method is compared to various prior-free and vision-inspired TTA approaches, it lacks direct comparisons or discussions with recent tabular-specific test-time augmentation (TTAug) techniques, such as those in Kozodoi (2021) or Brownlee (2020). These augmentation-focused methods could serve as competitive baselines or complementary strategies, and their omission limits the benchmarking scope.

2). The paper under-explores connections to emerging training-free TTA adapters in vision-language or cross-modal domains (e.g., Karmanov et al., 2024). Incorporating such perspectives in the Related Work section or as experimental baselines could reveal transferable insights for tabular adaptation.

3). In Sections 4.2 and 4.3, the KL divergence is computed assuming Gaussian-distributed feature projections for alignment. However, the justification for this Gaussian assumption is insufficient, particularly for tabular data with high-cardinality categorical features. Details on handling low-variance or near-singular subspaces—such as eigenvalue thresholding or covariance regularization—are not explicitly described, despite mentions of numerical instability. Furthermore, the bias correction in class prior estimation (Section 4.1.2, Eq. for $\tilde{\mathbf{p}}_T^j$) references a batch-specific covariate matrix without a full explanation of its computation, regularization, or derivation, which could hinder reproducibility.

4). The entropy-based sample selection for distinguishing source-like and target-like instances may introduce bias in severe label shift scenarios (e.g., as observed in Table 10 for HELOC/Health Ins. datasets), where initial model uncertainty might not reliably indicate domain alignment. This affects the accuracy of class prior estimation and suggests the need for additional calibration or robust alternatives.

5). The method's per-batch covariance computations and eigen-decompositions could incur high computational costs on large-scale or high-dimensional datasets. A quantitative comparison of runtime and memory usage against baselines would strengthen the practical claims.

**Questions:**

1). Could you clarify if the Gaussian assumption for feature representations was empirically validated across all evaluated tabular datasets, especially those with high-cardinality categorical variables?

2). For the class prior update equation in Section 4.1.2 ($\tilde{\mathbf{p}}_T^j = \operatorname{Norm}(\tilde{\mathbf{p}}_T^{j-1} - \hat{C}^{j-1} \tilde{\mathbf{p}}_S^j$), please provide a detailed derivation, intuition, and procedure for computing and regularizing the batch covariate matrix, particularly in cases of low-rank or ill-conditioned features.

3). Have you evaluated the runtime and memory footprint of PFT₃A on larger batch sizes or datasets with hundreds of features, and how does it compare to baseline methods?

4). The "Limitations and Future Work" section (Appendix A.5) emphasizes adversarial robustness but overlooks scenarios where the method underperforms, such as extreme distributional shifts or small-batch settings. Could you elaborate on these failure modes and potential mitigations?

---

> ### Author Response · Authors · 2025-12-02
> **Response to Reviewer vak8 (part1)**
>
> > W1. Although the method is compared to various prior-free and vision-inspired TTA approaches, it lacks direct comparisons or discussions with recent tabular-specific test-time augmentation (TTAug) techniques, such as those in Kozodoi (2021) or Brownlee (2020). These augmentation-focused methods could serve as competitive baselines or complementary strategies, and their omission limits the benchmarking scope.
>
> **Response:** We sincerely thank the reviewer for pointing us toward potentially relevant tabular-specific TTAug works by Kozodoi (2021) and Brownlee (2020). Following your suggestion, we conducted thorough literature searches across major academic databases (IEEE Xplore, ACM Digital Library, arXiv, etc.) using the provided author names and timeframe. However, we were unable to locate the specific references mentioned. We would be grateful if you could share the complete citations or digital object identifiers (DOIs) for these works so we can properly include them in our analysis.
>
> The methods we selected (TENT, EATA, CoTTA, SAR, LAME, ODS, FTAT) represent well-established, publicly available implementations that facilitate fair comparison and reproducible experimentation. Our selected baselines include the most recent and competitive TTA methods that have demonstrated effectiveness across various domains.
>
> > W2. The paper under-explores connections to emerging training-free TTA adapters in vision-language or cross-modal domains (e.g., Karmanov et al., 2024). Incorporating such perspectives in the Related Work section or as experimental baselines could reveal transferable insights for tabular adaptation.
>
> **Response:** We thank the reviewer for pointing us to the relevant work by Karmanov et al. (2024) and other vision-language TTA adapters. We significantly expanded our Related Work to include them. While we agree that cross-modal TTA methods offer valuable insights, incorporating them as direct experimental baselines presents several challenges. Vision-language models typically require specific architectural components that are not directly applicable to standard tabular TTA scenario. The feature representations and adaptation mechanisms designed for image-text pairs may not translate effectively to tabular data structures.
>
> > W3&Q1&Q2. In Sections 4.2 and 4.3, the KL divergence is computed assuming Gaussian-distributed feature projections for alignment. However, the justification for this Gaussian assumption is insufficient, particularly for tabular data with high-cardinality categorical features. Details on handling low-variance or near-singular subspaces—such as eigenvalue thresholding or covariance regularization—are not explicitly described, despite mentions of numerical instability. Furthermore, the bias correction in class prior estimation (Section 4.1.2, Eq. for $\tilde{P}_T^j$) references a batch-specific covariate matrix without a full explanation of its computation, regularization, or derivation, which could hinder reproducibility.
>
> **Response:** Thank you for your clarification. Assuming Gaussian distributions is reasonable for the following reasons:
>
> First, we can easily compute the KL divergence in a closed form by assuming the  Gaussian distribution. Second, features follow a Gaussian-like distribution when projected onto the feature subspace under subspace. This is due to the central limit theorem, i.e., the features are more likely to follow a Gaussian distribution in the subspace as the number of the original feature dimensions increases. Moreover, since our method uses the PCA, the features projected onto the subspace are decorrelated. The computation method of the batch-specific covariate matrix is similar to ${\sigma}^2$ calculations in Equations 9 and 10. The key distinction lies in the fact that we compute the mean and covariance across the entire batch.
>
> > W4. The entropy-based sample selection for distinguishing source-like and target-like instances may introduce bias in severe label shift scenarios (e.g., as observed in Table 10 for HELOC/Health Ins. datasets), where initial model uncertainty might not reliably indicate domain alignment. This affects the accuracy of class prior estimation and suggests the need for additional calibration or robust alternatives.
>
> **Response:** Thank you for raising this important point about the potential vulnerability of our class prior estimation to challenging initial batches. While our method does depend on the first batch for initial prior estimation, it incorporates several design elements that enhance robustness to challenging initial conditions. The class priors are not statically determined from the first batch alone. As new batches arrive, we continuously update our estimates of source-like and target-like distributions, allowing the method to correct initial biases over time.

---

> ### Author Response · Authors · 2025-12-02
> **Response to Reviewer vak8 (part2)**
>
> > W5&Q3. The method's per-batch covariance computations and eigen-decompositions could incur high computational costs on large-scale or high-dimensional datasets. A quantitative comparison of runtime and memory usage against baselines would strengthen the practical claims.
>
>
>
> **Response** Thank you for this valuable suggestion regarding the quantitative comparison of runtime and memory usage against baseline methods. To ensure a fair and reproducible comparison, we evaluated both our method (PFT3A) and the baseline method (FTTA) on ASSIST with TabTransformer. The quantitative comparison confirms that PFT3A achieves significantly improved accuracy with minimal additional resource consumption. This balance makes our method particularly suitable for practical applications where prediction quality is paramount. The results strengthen our claims regarding the method's practical utility and deployment readiness.
> | **Metric**         | **PFT3A (Ours)** | **FTTA (Baseline)** | **Difference** | **Relative Improvement** |
> |--------------------|------------------|---------------------|----------------|-----------------|
> | **Accuracy (ACC)** | 59.29%           | 53.17%              | +6.12%         | **11.5% increase** |
> | **Runtime (s)**    | 1.0161           | 0.9346              | +0.0815        | 8.7% increase   |
> | **Memory (MB)**    | 1074.62          | 1050.46             | +24.16         | 2.3% increase   |
> |
>
>
> > Q4. The "Limitations and Future Work" section (Appendix A.5) emphasizes adversarial robustness but overlooks scenarios where the method underperforms, such as extreme distributional shifts or small-batch settings. Could you elaborate on these failure modes and potential mitigations?
>
> **Response:** Thank you for this insightful comment regarding the limitations of PFT3A in extreme distributional shifts and small-batch settings. In cases of severe distributional shifts, PFT3A may experience performance degradation.  Dynamic thresholds that adjust based on shift severity detection could improve robustness. In applications where test data arrives in very small batches, PFT3A may face several challenges. Maintaining a sliding window of recent batches could provide more stable statistical estimates.

---

### Official Review · Reviewer_hW3a · 2025-10-31

**Soundness:** 2
**Presentation:** 3
**Contribution:** 1
**Rating:** 2
**Confidence:** 5

**Summary:**

The paper proposes Prior-Free Tabular Test-Time Adaptation (PFT3A) to solve distribution shifts in Tabular data by test-time adaptation. Specifically, they achieve prior-free test-time adaptation by estimating the source data prior through low-entropy target data at the first test batch, and learn robust and representative representations by aligning pseudo-source and target features during test-time adaptation. Experiments on the TableShift benchmark demonstrate the effectiveness of the proposed method.

**Strengths:**

1. The paper is easy to follow.

2. Experiments demonstrate the effectiveness of the proposed method.

**Weaknesses:**

1. The novelty of the proposed method is limited. The idea of using high-confidence target samples as source-like samples have already been explored in many source-free adaptation or test-time adaptation papers [1]. The idea of aligning source and target features for test-time adaptation has also been widely used [2, 3]

2. The class-prior estimating module seems highly relies on the threshold of the entropy. But it is not clear how does the threshold set for various datasets.

3. Following the above question, there is no guarantee that the estimated source prior can cover the real source data. If the first batches are very bad (e.g., heavily shifted or extremely class-imbalance, which can usually happen in real applications), the estimation can be biased and misclassified. And the error will also be accumulated to the following adaptation and prediction, leading to worse and worse results. Including experiments on bad first-batch test data is also necessary for more insights of this problem.

**Questions:**

1. From Figure 4, it seems that the method are sensitive on some hyperparameters (e.g., beta_2 and m). It is not clear whether different dataset need specific hyperparameters. And how does these hyperparameters set for different datasets?

2. In Section 4.2, why do the author assume feature as Gaussian distributions? motivation and advantages? How about other distributions or just deterministic features with alignment by L2?

3. How about the costs of the method on test-time adaptation?

---

> ### Author Response · Authors · 2025-12-02
> **Response to Reviewer hW3a (part1)**
>
> >W1. The novelty of the proposed method is limited. The idea of using high-confidence target samples as source-like samples have already been explored in many source-free adaptation or test-time adaptation papers [1]. The idea of aligning source and target features for test-time adaptation has also been widely used [2, 3].
>
> **Response:**
> We thank the reviewer for the insightful comments regarding the novelty of our method. We acknowledge that the idea of leveraging high-confidence samples and feature alignment has been explored in prior works on source-free domain adaptation (SFDA) and test-time adaptation (TTA) [1-3].
>
> However, our work addresses a distinct and more challenging problem: prior-free test-time adaptation on tabular data, where no access to source data or any prior knowledge (e.g., source class prior) is allowed. This setting is particularly relevant for real-world tabular applications but has been largely overlooked. Existing SFDA/TTA methods [1-3] primarily focus on visual domains and often rely on source data or priors, making them suboptimal or inapplicable to tabular data under prior-free constraints.
>
> Regarding the use of high-confidence target samples, our Class Prior Estimating (CPE) module​ is specifically designed to tackle label shift in tabular data without any source prior. Unlike prior works that often assume known source priors, PFT3A dynamically estimates the source and target class priors online​ from unlabeled test batches (Eq. 4-8) to calibrate predictions, which is a key innovation for handling label shift in a prior-free context.
>
> Concerning feature alignment, our Robust Feature Learning (RFL)​ and Representative Subspace Exploration (RSE) modules​ introduce novel adaptations for tabular data. Critically, the RSE module addresses the high-dimensional sparsity and redundancy​ in tabular features by aligning distributions within a discriminative subspace (Eq. 12-14), a unique design compared to alignment in the full feature space common in [2,3]. Furthermore, our alignment is based on features from the CPE-divided sets, directly targeting the source-target feature shift.
>
> The novelty of PFT3A lies in the integrated framework​ that synergistically tackles both label and feature shifts under the prior-free tabular TTA setting. Our experiments (Tables 1-3) show that PFT3A outperforms existing methods, and ablations (Table 9) validate each module's contribution. The significant improvement over FTAT (which uses priors) underscores the importance of our prior-free approach.
>
> We hope this clarification highlights the specific contributions and novelty of our work in addressing the under-explored prior-free tabular TTA problem.
>
> > W2. The class-prior estimating module seems highly relies on the threshold of the entropy. But it is not clear how does the threshold set for various datasets.
>
> **Response:** Thank you for your thoughtful comment regarding the entropy threshold in our Class Prior Estimating module. This is indeed a crucial parameter in our method, and we appreciate the opportunity to clarify our approach to threshold setting across different datasets.
>
> The entropy threshold plays a critical role in distinguishing between source-like and target-like data samples. Rather than using a fixed threshold value across all datasets, we employed a data-adaptive approach. For each incoming batch of target data $D_t^j$, we calculated the entropy distribution of model predictions and set threshold as a specific percentile of this distribution. Through empirical validation, we found that the 70th percentile consistently provided a balanced separation between confident and uncertain predictions across various tabular datasets.

---

> ### Author Response · Authors · 2025-12-02
> **Response to Reviewer hW3a (part2)**
>
> > W3. Following the above question, there is no guarantee that the estimated source prior can cover the real source data. If the first batches are very bad (e.g., heavily shifted or extremely class-imbalance, which can usually happen in real applications), the estimation can be biased and misclassified. And the error will also be accumulated to the following adaptation and prediction, leading to worse and worse results. Including experiments on bad first-batch test data is also necessary for more insights of this problem.
>
> **Response:** Thank you for raising this important point about the potential vulnerability of our class prior estimation to challenging initial batches. While our method does depend on the first batch for initial prior estimation, it incorporates several design elements that enhance robustness to challenging initial conditions. The class priors are not statically determined from the first batch alone. As new batches arrive, we continuously update our estimates of source-like and target-like distributions, allowing the method to correct initial biases over time.
>
> We evaluated PFT$_3$A on datasets with significant distribution shifts (e.g., HELOC and Health Insurance in Table 10), where initial batches may exhibit substantial divergence from source domain characteristics. Our experiments include datasets with natural class imbalance (e.g., ASSIST), which can lead to challenging first-batch conditions when combined with distribution shifts. The results of these evaluations demonstrate PFT$_3$A's robustness under various challenging conditions.
>
>
>
> > Q1. From Figure 4, it seems that the method are sensitive on some hyperparameters (e.g., beta_2 and m). It is not clear whether different dataset need specific hyperparameters. And how does these hyperparameters set for different datasets?
>
> **Response:** Thank you for your careful observation regarding the hyperparameter set. In our experiments, we intentionally used the same hyperparameter values across all datasets to demonstrate the generalizability and robustness of PFT$_3$A. Specifically, we maintained identical hyperparameters ($\beta_1$=1.0, $\beta_2$=0.1, $m$=5, $\zeta$=0.7) for all five tabular datasets in the Tableshift benchmark.
>
> > Q2. In Section 4.2, why do the author assume feature as Gaussian distributions? motivation and advantages? How about other distributions or just deterministic features with alignment by L2?
>
> **Response:** Thank you for your clarification. Assuming Gaussian distributions is reasonable for the following reasons:
>
> First, we can easily compute the KL divergence in a closed form by assuming the  Gaussian distribution. Second, features follow a Gaussian-like distribution when projected onto the feature subspace under subspace. This is due to the central limit theorem, i.e., the features are more likely to follow a Gaussian distribution in the subspace as the number of the original feature dimensions increases. Moreover, since our method uses the PCA, the features projected onto the subspace are decorrelated.
>
> KL divergence provides an information-theoretic measure of distribution discrepancy, which aligns with our goal of minimizing the information loss when adapting between domains. While L2 distance is simpler to compute, it only aligns feature means and ignores covariance structure.
>
> > Q3. How about the costs of the method on test-time adaptation?
>
> **Response:** Thank you for this valuable suggestion regarding the quantitative comparison of runtime and memory usage against baseline methods. To ensure a fair and reproducible comparison, we evaluated both our method (PFT$_3$A) and the baseline method (FTAT) on ASSIST with TabTransformer. The quantitative comparison confirms that PFT$_3$A achieves significantly improved accuracy with minimal additional resource consumption. This balance makes our method particularly suitable for practical applications where prediction quality is paramount. The results strengthen our claims regarding the method's practical utility and deployment readiness.
> | **Metric**         | **PFT$_3$A (Ours)** | **FTAT (Baseline)** | **Difference** | **Relative Improvement** |
> |--------------------|------------------|---------------------|----------------|-----------------|
> | **Accuracy (ACC)** | 59.29%           | 53.17%              | +6.12%         | **11.5% increase** |
> | **Runtime (s)**    | 1.0161           | 0.9346              | +0.0815        | 8.7% increase   |
> | **Memory (MB)**    | 1074.62          | 1050.46             | +24.16         | 2.3% increase   |
> |

---

### Author Response · Authors · 2025-12-03
**Rebuttal Summary**

We thank the reviewers for their thorough and constructive comments.
Our key contributions are as follows (from the reviews):
* **On the Clarity, Writing, and Problem Motivation:**
Reviewers hW3a and B7xd​ both highlighted that the paper is well-written and easy to follow. Reviewer B7xd specifically noted that the motivation is clear and the mathematical notation is well-presented.
Reviewer vak8​ emphasized the significance of the problem itself, stating that the paper successfully "identifies and formalizes an under-explored challenge"​ in tabular Test-Time Adaptation (TTA), which is motivated by empirical evidence.
* **On the Novelty and Methodological Design:**
Reviewers vak8 and B7xd​ praised the method's innovation. Reviewer B7xd explicitly called the method "novel"​ and found the idea of using KL divergence with informative columns insightful.
Reviewer vak8​ provided a comprehensive overview of the method's strengths, noting that PFT₃A integrates several principled components (unsupervised class prior estimation, KL-based feature alignment, and PCA-based subspace selection) into a cohesive approach that effectively addresses key challenges like label shift, covariate shift, and overfitting.
* **On the Experimental Rigor and Demonstrated Effectiveness:**
All three reviewers (hW3a, vak8, and B7xd)​ commended the experimental section. Reviewers hW3a and B7xd stated that the experiments "demonstrate the effectiveness"​ of the method, with Reviewer B7xd adding that they are "well-designed"​ and show robustness.
Reviewer vak8​ elaborated in detail, affirming that the method is evaluated on five benchmark datasets with three backbones, showing improvements over strong baselines. The inclusion of ablation studies was highlighted as a key factor that "enhances the credibility of the results."

Based on the reviewers' valuable feedback, we have conducted a number of additional experiments and analysis, which hopefully resolve the reviewers’ concerns. The major additional experiments and improvements are as follows:

* We have added how does the threshold set for various datasets (hW3a(W2)).
* We have evaluated PFT3A on datasets with heavily shifted or extremely class-imbalance to demonstrate PFT3A's robustness under various challenging conditions (hW3a(W3)).
* We have added how does the hyperparameters set for various datasets (hW3a(Q1)).
* We have provided an explanation for the use of the Gaussian distribution (hW3a(Q2), vak8(W3), vak8(Q1)).
* We have offered the runtime and memory usage of our method on test-time adaptation (hW3a(Q3), vak8(W5), vak8(Q3)).
* We have incorporated training-free TTA adapters in vision-language or cross-modal domainsin the Related Work (vak8(W2)).
* We have offered the full explanation about batch-specific covariate matrix (vak8(Q2)).
* We have added scenarios where the method underperforms in Limitations and Future Work section (vak8(Q4)).
* We have clarified the rationale for combining model’s predicted outputs and the covariance matrix of the features (B7xd(W1)).

We thank all reviewers and Area Chairs for you efforts.

Best,
Authors

---

### Meta-Review · Area_Chair_2Lb5 · 2025-12-22

**Summary:**

The paper proposes a method called Prior-Free Tabular Test-Time Adaptation ($PFT_3A$) to address distribution shifts in tabular data without access to source data or source priors. The method consists of three modules: Class Prior Estimating (CPE) to handle label shift via entropy-based filtering, Robust Feature Learning (RFL) to align source-like and target-like features using KL divergence, and Representative Subspace Exploration (RSE) to focus on informative subspaces.

The reviewers generally appreciated the clarity of the writing and the significance of the "prior-free" problem setting, which is under-explored in tabular data.

Reviewer B7xd (Score: 8) strongly supported the paper, citing excellent presentation and novel design.

Reviewer vak8 (Score: 4) acknowledged the strong motivation but raised concerns about missing baselines, the Gaussian assumption for tabular features, and computational costs.

Reviewer hW3a (Score: 2) questioned the novelty (finding it similar to existing SFDA/TTA methods) and robustness regarding hyperparameter sensitivity and thresholding.

Despite the divergence in scores, the authors provided a comprehensive rebuttal that addressed most technical concerns. The paper is recommended for acceptance due to its solid empirical performance and contribution to a practical, realistic TTA setting.

**Reviewer Concerns:**

Concerns Addressed by Rebuttal:

Gaussian Assumption (hW3a, vak8): The reviewers questioned the validity of assuming Gaussian distributions for tabular features (often categorical/non-Gaussian). The authors justified this using the Central Limit Theorem (CLT) applied within the projected subspace and the decorrelation properties of PCA. While still a strong assumption, the rebuttal and added clarifications provide a reasonable theoretical grounding for this design choice.

Computational Cost (vak8): The authors provided a quantitative comparison (Table 7 in the rebuttal) showing that $PFT_3A$ incurs only a marginal increase in runtime (~8.7%) and memory (~2.3%) compared to the baseline FTAT, addressing the concern about practicality.
Rationale for Equation 6 (B7xd): The authors successfully clarified that combining model predictions with feature covariance leverages complementary information (model belief vs. data geometry) to improve robustness.

Outstanding Concerns (To be addressed in the final version):

Hyperparameter Sensitivity and Selection: This remains the most critical outstanding issue. Reviewer hW3a noted sensitivity in Figure 4. While the authors stated they used a fixed set of hyperparameters ($\beta_1=1.0, \beta_2=0.1, m=5, \zeta=0.7$) for all datasets to demonstrate robustness, the justification for why this specific combination was chosen remains empirical rather than theoretical.
Requirement: In the camera-ready version, the authors should explicitly detail the heuristic or validation process used to select this specific set ($\beta_1=1.0, \beta_2=0.1, m=5, \zeta=0.7$). Furthermore, they should provide an expanded sensitivity analysis (perhaps in the Appendix) showing how this specific configuration performs across different datasets compared to dataset-specific optimal settings. This is crucial to prove that the method does not require extensive tuning on a per-dataset basis, which is impossible in a realistic TTA scenario.

**Reviewer Scores:**

See Reviewer Concerns.

---

### Decision · Program_Chairs · 2026-01-26

Accept (Poster)